# Soft ascent-descent as a stable and flexible alternative to flooding

**Matthew J. Holland**[*]
Osaka University

**Kosuke Nakatani**
Osaka University

## Abstract

As a heuristic for improving test accuracy in classification, the "flooding" method proposed by Ishida et al. (2020) sets a threshold for the average surrogate loss at training time; above the threshold, gradient descent is run as usual, but below the threshold, a switch to gradient *ascent* is made. While setting the threshold is non-trivial and is usually done with validation data, this simple technique has proved remarkably effective in terms of accuracy. On the other hand, what if we are also interested in other metrics such as model complexity or average surrogate loss at test time? As an attempt to achieve better overall performance with less fine-tuning, we propose a softened, pointwise mechanism called SoftAD (soft ascent-descent) that downweights points on the borderline, limits the effects of outliers, and retains the ascent-descent effect of flooding, with no additional computational overhead. We contrast formal stationarity guarantees with those for flooding, and empirically demonstrate how SoftAD can realize classification accuracy competitive with flooding (and the more expensive alternative SAM) while enjoying a much smaller loss generalization gap and model norm.

## 1 Introduction

Modern machine learning makes use of sophisticated models that are trained through optimization of non-convex objective functions, which typically admit numerous local minima that make for natural candidates when taken at face value. While many such candidates are indeed essentially "optimal" from the viewpoint of classification error rates or other average losses incurred at training time, these often turn out to be highly sub-optimal in terms of *performance at test time*. It goes without saying that understanding and closing this gap is the problem of "generalization" that underlies most machine learning research (Jiang et al., 2020; Dziugaite et al., 2020; Johnson and Zhang, 2023).

When we are faced with multiple candidates which are essentially optimal and thus indistinguishable in terms of some "base" objective function (e.g., the average loss) at training time, one of best-known heuristics for identifying good candidates is that of the "landscape" or "geometry" of the base objective in a neighborhood around each candidate. Roughly speaking, one expects that candidates in regions which are in some sense "flat" (often said to be less "sharp") tend to perform better at test time. Strictly speaking, flatness is not necessary for generalization (Dinh et al., 2017), but our intuition can often be empirically verified to be correct, as good generalization is regularly observed in flat regions where the eigenvalues of the Hessian are mostly concentrated near zero (Chaudhari et al., 2017). The spectral density of the Hessian can in principle be used to evaluate sharpness, and has well-known links to norms that can be used for explicit regularization (Karakida et al., 2019), but for large-scale neural network training in practice, first-order approximations have shown the greatest utility. In particular, the sharpness-aware minimization (SAM) algorithm of Foret et al. (2021), extended for scale invariance by Kwon et al. (2021) and later captured as a special case of the gradient norm penalization (GNP) scheme of Zhao et al. (2022), has shown state-of-the-art

---

[*]Corresponding author.

performance on a variety of deep learning tasks. All of these first-order procedures can be cast as (forward) finite-difference approximations of the curvature (Karakida et al., 2023), requiring at least double the computational cost of vanilla gradient descent (GD) at each iteration.

As an alternative approach, the "flooding" technique of Ishida et al. (2020) is worthy of attention for surprising improvements in test accuracy despite its apparent simplicity. Flooding is done as follows: fix a threshold $\theta$ before training and run vanilla GD until the average loss goes below $\theta$, and while below this threshold, run gradient *ascent* rather than descent (see §2.1 for details). Flooding appeared before SAM in the literature, but near the threshold, flooding can iterate between optimizing the empirical risk and the squared gradient norm (penalizing sharpness), establishing the former as an inexpensive alternative to the latter. On the other hand, it is not at all obvious how the flooding threshold $\theta$ should be set given a particular data distribution and model class, and at present there is no methodology for settings which are "optimal" or at least "sufficient" from the viewpoint of test accuracy. More importantly, what if we are interested in performance criteria going beyond that of classification accuracy? Flooding just says "make the average loss as close to $\theta$ as possible," and we hypothesize that this requirement is too weak to encourage low model complexity and/or good generalization in terms of losses, while also keeping test accuracy high.

In this work, we investigate the validity of this hypothesis, and consider the impact of making a stronger requirement, namely to ask the algorithm to "make sure the loss distribution is *well-concentrated* near $\theta$." We show in §3 that this can be implemented by introducing a smooth wrapper, applicable to any loss, which penalizes both over-performing and under-performing examples in a per-point fashion, instead of applying a hard threshold to the whole (mini-)batch as in flooding. We call this proposed procedure "soft ascent-decent" (SoftAD), and provide a detailed comparison with the flooding technique, highlighting the smoothness of SoftAD with implications in terms of formal stationarity guarantees, and emphasize how our mechanism leads to update directions that are qualitatively distinct from those used in flooding. Through rigorous empirical tests using both simulated and real-world benchmark classification datasets, featuring neural networks both large and small, we discover that compared with ERM, SAM, and flooding, the proposed SoftAD achieves far and away the smallest generalization error in terms of the base loss, while maintaining competitive accuracy and small model norms, without any explicit regularization.

Before diving into the main results just described, we introduce notation and background concepts in §2. SoftAD is introduced in §3, where we make basic empirical comparisons to flooding in §3.1, and contrast formal guarantees of stationarity for these two methods in §3.2. Our main empirical test results are given in §4, with discussion and concluding remarks wrapping up the paper in §5. All detailed proofs and supplementary empirical results are relegated to the appendix.

## 2 Background

To begin, we formulate performance metrics characterizing the problem of interest. Let $\mathcal{W} \subseteq \mathbb{R}^d$ parameterize our hypothesis class, let $\mathcal{Z}$ denote the set to which individual data points $z$ belong. Let $\mathsf{Z}$ represent a random (test) data point with distribution $\mu$ over $\mathcal{Z}$. For classification, where our data takes the form $\mathsf{Z} = (\mathsf{X}, \mathsf{Y})$ and for each $w \in \mathcal{W}$ we let $\widehat{\mathsf{Y}}(w)$ denote the predicted label given $\mathsf{X}$, the traditional performance metric of interest is the error probability at test time, denoted by

$$\mathcal{E}(w) := \mathbf{P}\left\{\widehat{\mathsf{Y}}(w) \neq \mathsf{Y}\right\}. \tag{1}$$

When we refer to test *accuracy*, we mean the probability of correct prediction, namely $1 - \mathcal{E}(w)$. Even when high accuracy is desired, it is standard to make use of computationally congenial surrogate loss functions for training. Let $\ell : \mathbb{R}^d \times \mathcal{Z} \to \mathbb{R}$ denote a generic loss function to be used for training. For example, if $z = (x, y)$ represents (input, label) pairs, then a typical choice of $\ell$ would be the cross-entropy loss. While the model is left abstract in our notation, note that for non-linear models such as neural networks, the mapping $w \mapsto \ell(w; z)$ may be non-convex and non-smooth over $\mathcal{W}$. As with the error probability (1), the expected loss (or "risk")

$$\mathrm{R}_\mu(w) := \mathbf{E}_\mu \, \ell(w; \mathsf{Z}) \tag{2}$$

is also an important indicator of classifier performance. Both (1) and (2) are ideal quantities since $\mu$ is unknown and $\mathsf{Z}$ is not observed at training time. We assume the learning algorithm has access to an independent sample of $n$ observations from $\mu$, denoted by $\mathbf{Z}_n := (\mathsf{Z}_1, \ldots, \mathsf{Z}_n)$ for

convenience. Traditional machine learning algorithms are driven by the empirical risk, denoted here by $R_n(w) := (1/n) \sum_{i=1}^{n} \ell(w; Z_i)$, in that they seek out (local) minima of $R_n$. In this paper, we use the term *empirical risk minimization* (*ERM*) to refer to algorithms that directly apply an optimizer to $R_n$. Under sophisticated models, the usual ERM objective $R_n(\cdot)$ tends to admit a complex landscape. As discussed in §1, numerous alternatives to ERM have been proposed, with the aim of minimizing $\mathcal{E}(\cdot)$ more reliably; next we take a closer look at one such technique, called "flooding."

## 2.1 Flooding

The basic intuition underlying the proposal of Ishida et al. (2020) is that while minimizing $R_n(\cdot)$ may be sufficient for maximizing the training accuracy, it need not be *necessary*, and from the perspective of optimizing $\mathcal{E}(\cdot)$, it may be worth it to sacrifice $R_n(\cdot)$ and even $R_\mu(\cdot)$. Fixing a threshold $\theta \in \mathbb{R}$, the "flooded" objective is

$$F_n(w; \theta) := \theta + |R_n(w) - \theta|. \tag{3}$$

This objective can be implemented as a simple wrapper around typical loss functions, to which off-the-shelf gradient-based optimizers can be applied; running vanilla sub-gradient descent yields a sequence $(w_1, w_2, \ldots)$ generated using the update

$$w_{t+1} = w_t - \alpha \operatorname{sign}(R_n(w_t) - \theta) \nabla R_n(w_t) \tag{4}$$

for all $t \geq 1$, where $\operatorname{sign}(x) := x/|x|$ for all $x \neq 0$ and $\operatorname{sign}(0) := 0$, and $\alpha > 0$ is a fixed step size. The update (4) characterizes what we call the *Flooding* algorithm. From the above definitions, $F_n(w; \theta)$ is minimal if and only if $R_n(w) = \theta$. That is, Flooding seeks out $w$ such that the training loss distribution induced by $(\ell(w; Z_1), \ldots, \ell(w; Z_n))$ has a mean which is close to $\theta$; nothing more, nothing less.

## 2.2 Links to sharpness

Assuming for now that the loss is differentiable, it has been appreciated for some time that the *distribution* of the loss gradients $\nabla \ell(w; Z)$ can convey important information about generalization (Zhang et al., 2020), and in particular the role of gradient regularization, both implicit and explicit, is receiving significant attention (Barrett and Dherin, 2021; Smith et al., 2021).[2] As a concrete example of explicit regularization, consider modifying the ERM objective using the squared Euclidean norm as

$$\widetilde{R}_n(w; \lambda) := R_n(w) + \frac{\lambda}{2} \|\nabla R_n(w)\|^2 \tag{5}$$

where $\lambda \geq 0$ controls the degree of penalization. If one is to minimize this objective in $w$ directly using gradient descent, this involves computing

$$\nabla \widetilde{R}_n(w; \lambda) = \nabla R_n(w) + \lambda \nabla^2 R_n(w) (\nabla R_n(w))$$

and thus doing matrix multiplication using a $d \times d$ Hessian $\nabla^2 R_n(w)$, an unattractive proposition when $d$ is large. A linear approximation to the expensive term can be obtained via

$$\frac{\nabla R_n(w + au) - \nabla R_n(w)}{a} \approx \nabla^2 R_n(w)(u)$$

where $u \in \mathbb{R}^d$ is arbitrary and $|a|$ is small; see Zhao et al. (2022, §3.3) for example. Applying this to approximate $\nabla \widetilde{R}_n(w; \lambda)$, we have

$$\nabla R_n(w) + \frac{\lambda}{a} (\nabla R_n(w + a\nabla R_n(w)) - \nabla R_n(w)) \approx \nabla \widetilde{R}_n(w; \lambda). \tag{6}$$

The iterative update directions used by the *SAM* algorithm of Foret et al. (2021) are captured by setting $a = \lambda$, offering a nice link between loss-based sharpness control and gradient regularization. The extension of SAM in GNP (Zhao et al., 2022) can be expressed by an analogous derivation, replacing

---

[2]Throughout this paper, all gradients are taken with respect to $w \mapsto \ell(w; z)$, assumed to exist on an open set containing $\mathcal{W}$ for all $z \in \mathcal{Z}$. It should however be noted that gradients taken with respect to parts of the data $z$ have been used in objective function design for years; see Drucker and Le Cun (1992).

the squared norm $\|\cdot\|^2$ in (5) with $\|\cdot\|$. Using updates of the form given in (6) with $a > 0$ is called a "forward" finite-difference (FD) approach to explicit gradient regularization (GR) (henceforth, *FD-GR*), and clearly requires two gradient calls per update.[3] Better precision is available using "centered" FD, at the cost of additional gradient calls (Karakida et al., 2023). How does this all relate to Flooding? In repeating the update (4), once the empirical risk $R_n$ goes below $\theta$ and the algorithm switches from ascent to descent, it is straightforward to show conditions where this leads to iteration between minimizing $R_n$ and $\|\nabla R_n(w)\|^2$. We give more details in §B.1.

## 3    Soft ascent-descent

With the context of §2 in place, we consider making two qualitative changes to the Flooding update (4), described as follows.

1. **Pointwise thresholds:** Invert the order of applying $R_n(\cdot)$ and $\mathrm{sign}(\cdot)$, i.e., do summation over data points *after* per-loss truncation.

2. **Soft truncation:** Replace the hard threshold $\mathrm{sign}(\cdot)$ with a continuous, bounded, monotonic (increasing) function $\phi(\cdot)$ satisfying $\phi(0) = 0$.

The reason for making the thresholds pointwise is to allow the algorithm to view "ascent" and "descent" from the perspective of individual losses (rather than bundled up in $R_n$), making it possible to utilize a sum of both ascent and descent update directions.[4] To make this sum a *weighted* sum, the soft truncation using $\phi$ plays a key role. Keeping $\phi$ bounded limits the impact of errant loss values, while the other assumptions allow for both ascent and descent, with "borderline" points near the threshold given *less* weight. Written explicitly as an empirical objective function, we use

$$S_n(w; \theta) := \theta + \frac{1}{n} \sum_{i=1}^{n} \rho(\ell(w; Z_i) - \theta) \tag{7}$$

where once again $\theta \in \mathbb{R}$ is a fixed threshold, and we set $\rho(x) := \sqrt{x^2 + 1} - 1$. Running vanilla GD on $S_n(\cdot; \theta)$ in (7) yields the update

$$w_{t+1} = w_t - \frac{\alpha}{n} \sum_{i=1}^{n} \phi(\ell(w_t; Z_i) - \theta) \nabla \ell(w_t; Z_i), \tag{8}$$

with $\phi(x) := \rho'(x) = x/\sqrt{x^2 + 1}$, and $\alpha > 0$ is once again a fixed step size. For convenience, we use *soft ascent-descent* (or *SoftAD* for short) to refer to the algorithm implied by the iterative update (8). Note that there is nothing particularly special about our choice of $\rho$ here; it is just a simple algebraic function whose derivative also takes a simple form; note that the function $\phi$ resulting from our choice of $\rho$ satisfies the desired properties of continuity, boundedness, monotonicity, and $\phi(0) = \rho'(0) = 0$.[5] Note that for each point being summed over, $\ell(w_t; Z_i) > \theta$ implies descent while $\ell(w_t; Z_i) < \theta$ implies ascent, and borderline points with $\ell(w_t; Z_i) \approx \theta$ have a smaller relative impact. In contrast with Flooding, SoftAD requires that the training loss distribution induced by $(\ell(w; Z_1), \ldots, \ell(w; Z_n))$ be well-concentrated around $\theta$, where the degree of dispersion is measured in a symmetric fashion using $\rho$.

*Remark* 1 (Comparison with other variants of Flooding). During the review phase for this work, we were made aware of another recent related work by Xie et al. (2022). Their proposed method is known as *iFlood*, and it is essentially a middle ground between our proposal above and the original Flooding procedure. They use pointwise thresholds as we do in SoftAD, but retain the hard ascent-descent switch as in Flooding. More concisely, replacing our soft truncator $\phi(x)$ in (8) with the absolute value $|x|$ yields the iFlood update. We have added empirical test results for iFlood to complement our original experiments at the end of §C.2. Another very recent variant is *AdaFlood* (Bae et al., 2023), which sets the $\theta$ threshold level individually for each point based on "difficulty" as evaluated using an auxiliary model.

---

[3] In the current example, one call at $w$, and another call at $w + a\nabla R_n(w)$.

[4] This cannot be achieved by taking a mini-batch of size 1 when using Flooding, since the number of gradients summed over always equals the number of losses averaged when checking the ascent/descent condition.

[5] There are many other possible candidates, but this is typical "smooth Huber function," also known in the literature as pseudo-Huber, Charbonnier, and L1-L2 (Barron, 2019).

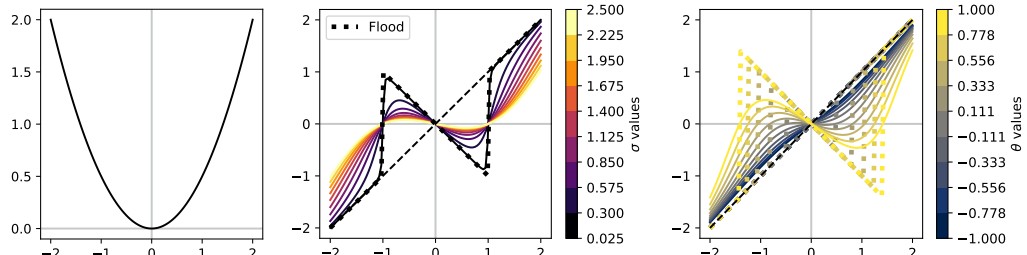

**Figure 1:** The left-most figure simply plots the graph of $f(x) = x^2/2$ over $x \in [-2, 2]$. The two remaining figures show plots of the graphs of $f'(x) = x$ (dashed black line) and $\phi((f(x) - \theta)/\sigma) f'(x)$ for the same range of $x$ values, with colors corresponding to modified values of $\sigma$ (middle plot; $\theta = 0.5$ fixed) and $\theta$ (right-most plot; $\sigma = 1.0$ fixed) respectively. Thick dotted lines are $\phi = \text{sign}$, thin solid lines are $\phi = \rho'$.

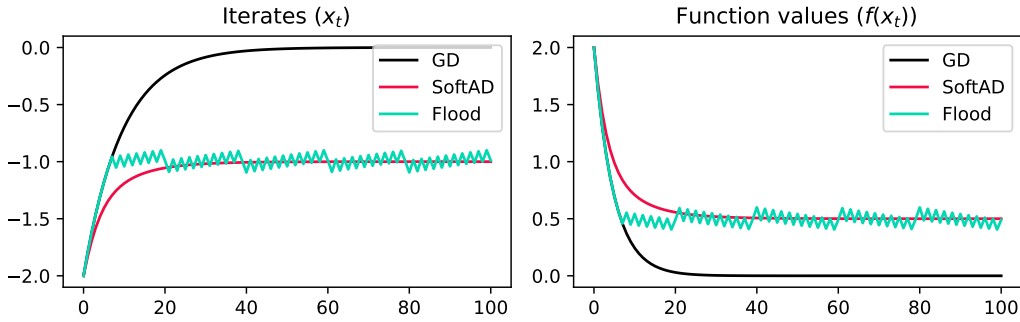

**Figure 2:** Gradient descent on the quadratic example from Figure 1. The horizontal axis denotes iteration number, and we plot sequences of iterates $(x_t)$ and function values $(f(x_t))$ for each method. Here "GD" denotes vanilla gradient descent, with "Flood" and "SoftAD" corresponding to (4) and (8) respectively. Step size is $\alpha = 0.1$.

*Remark* 2 (Difference from OCE-like criteria). At first glance, our objective $\mathsf{S}_n(w; \theta)$ in (7) may appear similar to the criteria used in OCE risk minimization (Lee et al., 2020; Li et al., 2021) and some varieties of DRO (Duchi and Namkoong, 2021). Aside from the obvious difference that $\theta$ is fixed, rather than optimized alongside $w$, the critical difference here is that our $\rho(\cdot)$ is *not* monotonic. Losses which are too large and too small are *both* penalized. It is precisely this bi-directional property that allows for switching between ascent and descent; this is impossible to achieve with monotonic OCE and DRO risks (Holland, 2023; Holland and Tanabe, 2023; Hu et al., 2023; Royset, 2024). This bi-directional criterion can also be used as a straightforward method to provably avoid unintentional "collapse" into ERM solutions (Holland, 2024).

### 3.1 Initial comparison with Flooding

To develop some intuition for our SoftAD (8) and the Flooding update (4), we carry out a few illustrative numerical experiments. To start, let us consider a simple, non-stochastic example in one dimension. Letting $f : \mathbb{R} \to \mathbb{R}$ be some differentiable function, we consider how the transformed gradient $\phi(f(x) - \theta) f'(x)$ behaves under $\phi = \text{sign}$ and $\phi = \rho'$. In Figure 1, we give a numerical example using a quadratic function. The softened nature of the transformed gradients used in SoftAD is clear when compared with the hard switching mechanism underlying the Flooding update. In Figure 2, we continue the quadratic function example, looking at sequences $(x_1, x_2, \ldots)$ generated based on the Flooding and SoftAD procedures. That is, instead of $n$ points from which to compute losses, we have just one "loss," namely $f(x) = x^2/2$. Both procedures realize an effective "flood level" of sorts (i.e., a buffer around the loss minimizer), but as expected, the Flooding procedure tends to be far more "jagged" in its trajectory.

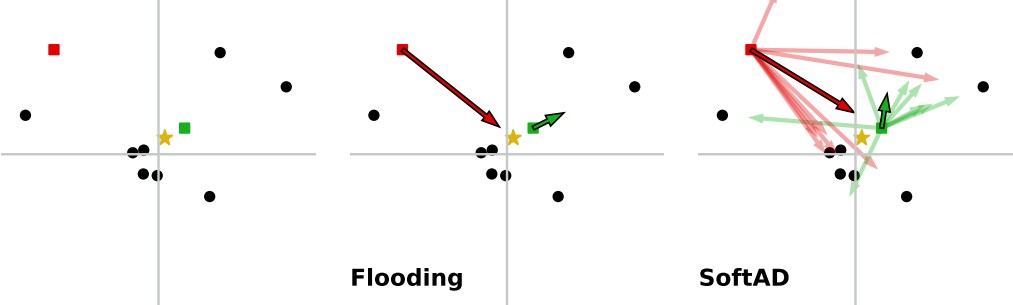

**Figure 3:** *Left:* We randomly sample $n = 8$ points (black dots) from the 2D Gaussian distribution, zero mean, zero correlations, with standard deviation $2\sqrt{2}$ in each coordinate. The two candidates are denoted by square-shaped points (red and green), and the minimizer of $\mathsf{R}_n$ is given by a gold star. *Center:* The Flooding updates (colored arrows) via (4) for each candidate. *Right:* Analogous SoftAD update vectors via (8), with per-point transformed gradients (semi-transparent arrows) for reference. Throughout, we have fixed $\theta = 1.5 \times \min_w \mathsf{R}_n(w)$ and $\alpha = 0.75$.

Finally, a simple example to illustrate how the per-point soft thresholding of SoftAD leads to distinct gradient-based update directions when compared to the Flooding strategy. Here we consider a dataset of $n$ data points $z_1, \ldots, z_n \in \mathbb{R}^2$, and use the squared Euclidean norm as a loss, i.e., $\ell(w; z) = \|w - z\|^2$. This is a natural extension of the quadratic example in the previous paragraph, to multiple points and two dimensions. In Figure 3, we look at two candidate points, and compute the Flooding and SoftAD update directions that arise at each candidate under a randomly generated dataset. We can clearly see how the Flooding update plunges directly towards the minimizer of $\mathsf{R}_n$, unless it is too close (given threshold $\theta$), in which case it goes in the opposite direction. In contrast, the SoftAD update is composed of per-point update directions, some which attract toward the minimum, and some which repel from the minimum, with borderline points down-weighted (shorter update arrows). Since the final update averages over these, movement both toward and away from the minimum is clearly "softened" when compared with the Flooding updates.

### 3.2 Comparison of convergence properties

With the Flooding and SoftAD methods in mind, next we consider concrete conditions under which stochastic gradient-based learning algorithms can be given guarantees of (approximate) stationarity. Throughout this section, we assume that the loss $w \mapsto \ell(w; z)$ is locally Lipschitz on $\mathcal{W}$ for each $z \in \mathcal{Z}$, but convexity will not be used. Let us assume for simplicity that $(\mathsf{Z}_1, \mathsf{Z}_2, \ldots)$ is a sequence of independent random variables with distribution $\mu$, the same as our test data point $\mathsf{Z} \sim \mu$.

Starting with SoftAD, we are interested in procedures fuelled by stochastic gradients of the form

$$\mathsf{G}_t(w) \coloneqq \phi(\ell(w; \mathsf{Z}_t) - \theta) \nabla \ell(w; \mathsf{Z}_t) \tag{9}$$

for all integers $t \geq 1$ and $w \in \mathcal{W}$, with threshold $\theta \in \mathbb{R}$ fixed in advance, and $\phi = \rho'$ as before. Recalling the empirical SoftAD objective $\mathsf{S}_n$ in (7), the underlying population objective is

$$\mathsf{S}_\mu(w) \coloneqq \theta + \mathbf{E}_\mu \, \rho(\ell(w; \mathsf{Z}) - \theta). \tag{10}$$

By design, we do not expect SoftAD to approach a stationary point of the original $\mathsf{R}_\mu$. Note that under mild assumptions on the data distribution, we have an unbiased estimator with $\mathbf{E}_\mu \, \mathsf{G}_t(w) = \nabla \mathsf{S}_\mu(w)$, suggesting stationarity in terms of $\mathsf{S}_\mu(\cdot)$ as a natural goal. Assuming the losses are $L_\ell$-smooth in expectation, one can readily confirm that the objective (10) is $L_{\mathrm{AD}}$-smooth, with

$$L_{\mathrm{AD}} \coloneqq \mathbf{E}_\mu \left[ \sup_{w \in \mathcal{W}} \|\nabla \ell(w; \mathsf{Z})\|^2 \right] + L_\ell. \tag{11}$$

A more detailed derivation is given in §B.5. Assuming that second-order moments are finite in a uniform sense over $\mathcal{W}$ ensures that $L_{\mathrm{AD}} < \infty$, and naturally implies pointwise variance bounds, allowing us to seek out stationarity guarantees using a combination of gradient norm control and momentum in the fashion of Cutkosky and Mehta (2021).

**Proposition 3** (Stationarity for SoftAD, smooth case). *Starting with an arbitrary $w_1 \in \mathcal{W}$, update using $w_{t+1} = w_t - \alpha \mathsf{M}_t / \|\mathsf{M}_t\|$, where $\mathsf{M}_t := b\mathsf{M}_{t-1} + (1-b)\bar{\mathsf{G}}_t(w_t)$ for $t \geq 1$, with $\mathsf{M}_0 := 0$ and $\bar{\mathsf{G}}_t(\cdot) := \mathsf{G}_t(\cdot) \min\{1, \gamma/\|\mathsf{G}_t(\cdot)\|\}$, taking each gradient $\mathsf{G}_t(\cdot)$ based on (9). Assuming we make $T - 1$ updates, set the momentum parameter to $b = 1 - 1/\sqrt{T}$, the norm threshold to $\gamma = \sqrt{(L_{\mathrm{AD}} - L_\ell)/(1-b)}$, and the step size to $\alpha = 1/T^{3/4}$. The stationarity of this sequence $(w_1, w_2, \ldots)$, assumed to be in $\mathcal{W}$, in terms of the modified objective (10) can be bounded by*

$$\frac{1}{T} \sum_{t=1}^{T} \|\nabla \mathsf{S}_\mu(w_t)\| \leq \frac{1}{T^{1/4}} \left( \mathsf{S}_\mu(w_1) - \mathsf{S}_\mu(w_{T+1}) + \frac{3L_{\mathrm{AD}}}{2} + 2\sqrt{L_{\mathrm{AD}} - L_\ell}\,(1 + C_\delta) \right)$$

*using confidence-dependent factor $C_\delta := 10\log(3T/\delta) + 4\sqrt{\log(3T/\delta)} + 1$, with probability no less than $1 - \delta$ over the draw of $(\mathsf{Z}_1, \mathsf{Z}_2, \ldots)$.*

For comparison, we next consider stationarity of the Flooding algorithm in the same context of smooth losses. The argument used in Proposition 3 critically depends on the smoothness of the underlying objective (10). Unfortunately, this smoothness cannot be leveraged when we consider the population objective underlying the Flooding procedure, namely the function

$$w \mapsto \theta + |\mathsf{R}_\mu(w) - \theta|. \tag{12}$$

Further complicating things is the fact that stochastic gradients of the form (9) with $\phi(\cdot)$ replaced by $\mathrm{sign}(\cdot)$ do not yield unbiased sub-gradient estimates for (12), but rather for an upper bound $\theta + \mathbf{E}_\mu|\ell(w; \mathsf{Z}) - \theta|$ that follows via Jensen's inequality. A lack of smoothness means we cannot establish stationarity in terms of (12) nor the upper bound just given, but it is possible using a smoothed approximation (the Moreau envelope) of this bound:

$$\bar{\mathsf{F}}_\mu(w) := \inf_{v \in \mathcal{W}} \left[ \theta + \mathbf{E}_\mu|\ell(v; \mathsf{Z}) - \theta| + \frac{1}{2\beta}\|v - w\|^2 \right]. \tag{13}$$

The parameter $\beta > 0$ controls the degree of smoothness. The objective in (13) can be linked to "gradients" in (9) with $\phi = \mathrm{sign}$, and leveraging the Lipschitz continuity of $|\cdot|$ along with a sufficiently smooth loss, it is possible to show that this non-smooth objective satisfies a weak form of convexity, and using the techniques of Davis and Drusvyatskiy (2019) it is possible to show that stochastic gradient algorithms enjoy stationarity guarantees, albeit not in terms of the objective (12), but rather the smoothed upper bound (13).

**Proposition 4** (Stationarity for Flooding, smooth case). *Letting $\mathcal{W}$ be closed and convex, take an initial point $w_1 \in \mathcal{W}$, and make $T - 1$ updates using $w_{t+1} = \Pi_{\mathcal{W}}[w_t - \alpha \mathsf{G}_t(w_t)]$, where $\Pi_{\mathcal{W}}[\cdot]$ denotes projection to $\mathcal{W}$, and each $\mathsf{G}_t(\cdot)$ is computed using (9) with $\phi = \mathrm{sign}$. Assuming the loss $w \mapsto \ell(w; z)$ is $L_\ell^*$-smooth on $\mathcal{W}$ for all $z \in \mathcal{Z}$, and taking a step size of*

$$\alpha^2 = \frac{\Delta}{TL_\ell^*(L_{\mathrm{AD}} - L_\ell)}, \ \textit{using } \Delta \textit{ such that } \Delta \geq \bar{\mathsf{F}}_\mu(w_1) - \inf_{w \in \mathcal{W}} \bar{\mathsf{F}}_\mu(w),$$

*with $L_{\mathrm{AD}}$ and $L_\ell$ as in (11), the expected squared stationarity in terms of the smoothed upper bound (13) at smoothness level $\beta = 1/(2L_\ell^*)$ can be controlled as*

$$\frac{1}{T} \sum_{t=1}^{T} \mathbf{E}\|\nabla \bar{\mathsf{F}}_\mu(w_t)\|^2 \leq \sqrt{\frac{2L_\ell^*(L_{\mathrm{AD}} - L_\ell)\Delta}{T}}$$

*with expectation taken over the draw of $(\mathsf{Z}_1, \mathsf{Z}_2, \ldots)$.*

*Remark* 5 (Comparing rates and assumptions). Considering the preceding Propositions 3 and 4, one common point is that learning algorithms based on both the SoftAD and Flooding gradients (of mini-batch size 1) can be shown to be approximately stationary in terms of functions of a similar form (i.e., (10) and (12)), differing only in how they measure deviations from the threshold $\theta$. The rates of decrease (as a function of $T$) are essentially the same, noting that the bounds in Proposition 4 are in terms of *squared* norms. That said, a lack of smoothness means the Flooding guarantees only hold for a smoothed variant, plus they require a stronger form of smoothness in the loss (over all $z$ vs. in expectation). In addition, the SoftAD guarantees hold with high probability over the data sample, and can be readily strengthened to hold for an individual iterate (instead of summing over $T$ iterates), using for example the technique of Cutkosky and Mehta (2021, Thm. 3).

|        | Gaussian | Sinusoid | Spiral | CIFAR-10 | CIFAR-100 | Fashion | SVHN  |
|--------|----------|----------|--------|----------|-----------|---------|-------|
| ERM    | 0.080    | 0.150    | 0.297  | 3.265    | 7.603     | 0.801   | 0.762 |
| Flood  | 0.011    | 0.058    | 0.119  | 1.239    | 3.114     | 0.436   | 0.436 |
| SAM    | 0.024    | 0.096    | 0.154  | 1.512    | 3.672     | 0.639   | 0.493 |
| SoftAD | **0.004** | **0.016** | **0.087** | **1.168** | **2.701** | **0.422** | **0.362** |

**Table 1:** Generalization gap (test - training) for trial-averaged cross entropy loss after final epoch.

## 4 Empirical study

In this section, we apply the proposed SoftAD procedure to a variety of classification tasks using neural network models, leading to losses that are non-convex and non-smooth. Our goal here is to compare and contrast the behavior and performance (accuracy, average loss, model norm) of SoftAD with three natural alternatives: ERM, Flooding, and SAM.[6]

### 4.1 Overview of experiments

Our core experiments are centered around re-creating the tests done by Ishida et al. (2020, §4.1, §4.2) and Foret et al. (2021, §3.1) to include all four methods of interest. There are two main parts: simulation-based tests and real benchmark-based tests. We briefly describe the setup of each below.

**Non-linear binary classification on the plane**  We use three synthetic data-generators ("two Gaussians," "sinusoid," and "spiral," see Figure 7) to create a dataset on the 2D plane that is not linearly separable, but separable using relatively simple non-linear models. We treat the underlying model as unknown, and approximate it using a shallow feedforward neural network. All four methods of interest (ERM, Flooding, SAM, and SoftAD) are driven by the Adam optimizer, with the cross-entropy loss used as the base loss function. Complete experimental details are provided in §C.1.

**Image classification from scratch**  Our second set of experiments utilizes four well-known benchmark datasets for multi-class image classification. Compared to the synthetic experiments, the classification task is more difficult (much larger inputs, variation within classes, more classes), and so we utilize more sophisticated neural network models to tackle the classification task. That said, as the sub-section title indicates, this training is done "from scratch," i.e., no pre-trained models are used. The datasets we use are all standard benchmarks in the machine learning community: CIFAR-10, CIFAR-100, FashionMNIST, and SVHN. Model choice essentially mirrors that of Ishida et al. (2020, §4.2). For FashionMNIST, we flatten each image into a vector, and use a simple feedforward neural network with one hidden layer. For SVHN, we use ResNet-18 as implemented in `torchvision.models`, without any pre-trained weights. Finally, for both CIFAR-10 and CIFAR-100, we use ResNet-34 (again in `torchvision.models`) without pre-training. For the optimizer, we use vanilla SGD with a fixed step size. Full details are given in §C.2 in the appendix.

### 4.2 Main findings

**Uniformly small loss generalization gap**  One of the most lucid results we obtained is that SoftAD shows the smallest *loss* generalization gap of all the methods studied, across all models and datasets used. In Table 1, we show the gaps incurred under each dataset. More precisely, for each trial and each epoch, we compute the average cross-entropy loss on test and training (less validation) datasets, and then respective average both of these over all trials. The difference of these two values (i.e., trial-averaged test loss minus trial-averaged training loss) after the final epoch of training is the value shown in each cell of the table.

**Balance of accuracy and loss on real data**  In the previous paragraph we noted that SoftAD has superior loss gaps, but this is not much to celebrate if performance in terms of the key metrics of interest (i.e., test loss and test accuracy) is poor. In Figures 4–5, we show the trajectory of loss and accuracy (for both training and test data) over epochs run (averaged over trials). All four methods

---

[6]To re-create all of the numerical test results and figures from this paper, source code and Jupyter notebooks are available at a public GitHub repository: `https://github.com/feedbackward/bdd-flood`.

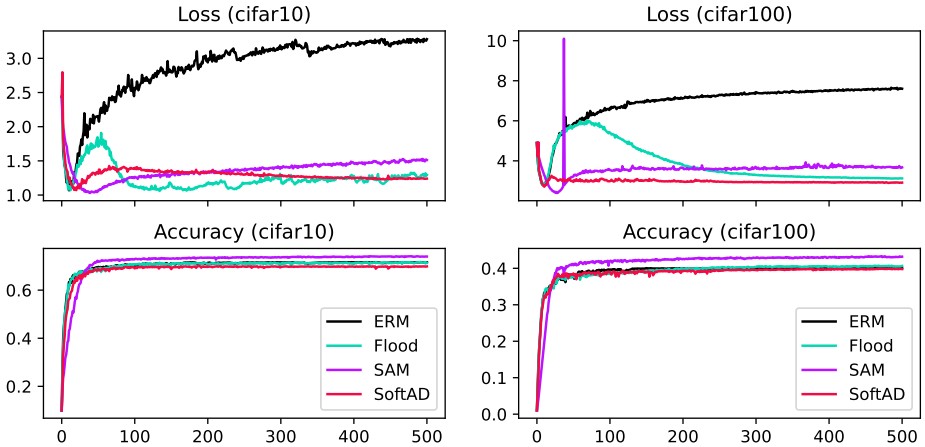

**Figure 4:** Trajectories over epochs for average test loss (top row) and test accuracy (bottom row). Horizontal axis is epoch number. Columns are associated with the CIFAR-10 and CIFAR-100 datasets (left to right).

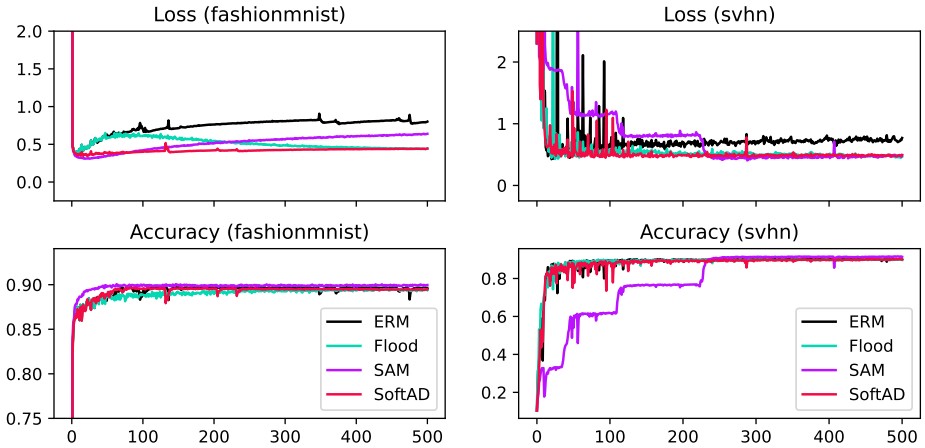

**Figure 5:** Analogous to Figure 4, but with FashionMNIST and SVHN datasets.

are comparable in terms of accuracy, with SAM (at double the gradient cost) coming out slightly ahead. On the other hand, there is significant divergence between the different methods in terms of test loss. For each dataset, SoftAD achieves a superior test loss, often converging faster than any of the other methods; this may be a natural by-product of the fact that SoftAD is designed to ensure losses are *well-concentrated* around threshold $\theta$, instead of just asking that their mean get close to $\theta$ (as in Flooding). While there is clearly a major difference between ERM and the other three methods, the stable nature of the "double descent" in SoftAD is quite stark compared with Flooding and SAM.

**Uniformly smaller model norms** We do not do any explicit model regularization (e.g., L2 norm penalization) in our experiments here, and we only use fixed step-size parameters for Adam and SGD, so as we run for many iterations, the norm of the model weight parameters tends to grow. While this property holds across all methods tested here, we find that under all datasets and models tested, SoftAD uniformly results in the smallest model norm; see Figure 6 for trajectories over epochs for each benchmark dataset. The "Model norm" values plotted here are the L2 norm of all the model parameters (neural network weights) concatenated into a single vector, and these norm values are averaged over trials. Completely analogous trends hold for the simulated datasets as well.

**Trends in hyperparameter selection** Aside from ERM, the three key methods of interest (Flooding, SAM, SoftAD) each have one hyperparameter. Flooding and SoftAD have the threshold $\theta$ as described

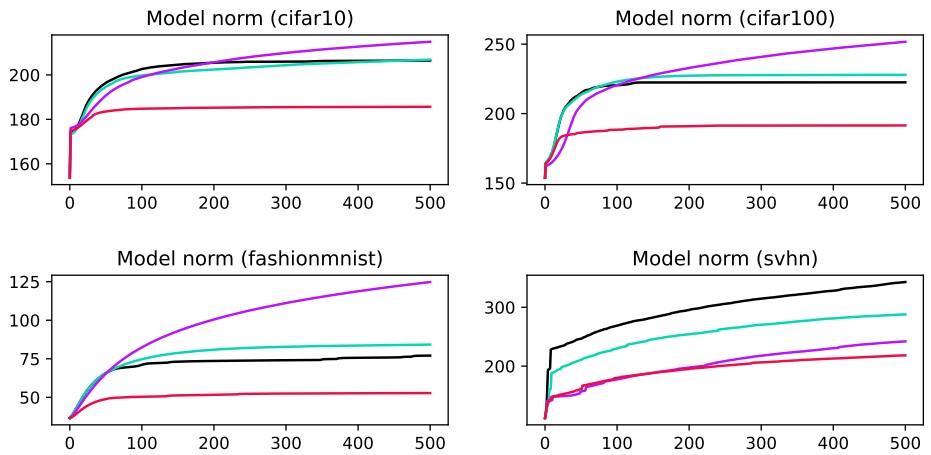

**Figure 6:** Model norm trajectories over epochs for each dataset in Figures 4–5.

|  | Gaussian | Sinusoid | Spiral | CIFAR-10 | CIFAR-100 | Fashion | SVHN |
|---|---|---|---|---|---|---|---|
| Flood | 0.05 (0.06) | 0.05 (0.06) | 0.05 (0.06) | 0.05 (0.06) | 0.01 (0) | 0.01 (0) | 0.03 (0.05) |
| SAM | 0.36 (0.19) | 0.05 (0.06) | 0.05 (0.06) | 0.36 (0.19) | 0.5 (0) | 0.32 (0.16) | 0.28 (0.20) |
| SoftAD | 0.08 (0.08) | 0.05 (0.06) | 0.05 (0.06) | 0.08 (0.08) | 0.22 (0.14) | 0.03 (0.04) | 0.13 (0.10) |

**Table 2:** Hyperparameters selected by validation for each method (averaged over trials). Flooding and SoftAD have threshold $\theta$; SAM has radius parameter. Standard deviation (over trials) is given in small-text parentheses.

in §2–§3, and SAM has the radius parameter (denoted by "$\rho$" in the original paper). In all tests, we select a representative candidate for each method with hyperparameters by using validation data held out from the training data, and in Table 2 we show the average and standard deviation of the validation-based hyperparameters over randomized trials (see §C for exact hyperparameter grid values). One clear take-away from this table is that the "best" value of $\theta$ (in terms of accuracy) for SoftAD tends to be *larger* than that for Flooding, and this trend is uniform across all datasets, both simulated and real. In particular for the real benchmark datasets, it is interesting to note that while a larger threshold $\theta$ (applied to *loss* distribution) is selected for SoftAD, the resulting test loss value achieved is actually smaller/better than that achieved by Flooding (top row of Figures 4-5).

## 5   Limitations and concluding remarks

While previous work had already shown that it is possible to sacrifice performance in terms of losses to improve accuracy, the nature of that tradeoff was left totally unexplored, and in §1 we put forward the hypothesis that simply asking the empirical loss mean to get close to a non-zero threshold $\theta$, as in Flooding, would not be enough to realize a competitive tradeoff over varied learning tasks (datasets, models). Our main take-away is that we have empirical evidence that the slightly stronger requirement of "*losses well-concentrated around $\theta$*" (implemented as SoftAD) can result in an appealing balance of average test loss and accuracy, with the added benefit of a strong (implicit) regularization effect, likely due to the soft dampening effect on borderline points. A more formal theoretical understanding of this regularization effect is of interest, as are empirical studies going far beyond the limited choice of loss functions used here. Our biggest limitation is that the question of "how to set the threshold $\theta$?" still remains without an answer. Any meaningful answer will likely require some user judgement regarding tradeoffs between performance metrics. One potential first approach would be to leverage recent techniques for estimating the Bayes error (Ishida et al., 2023), combined with existing surrogate theory (Bartlett et al., 2006) to reverse-engineer a loss threshold given a user-specified "tolerable" drop in accuracy, for example.

## Acknowledgments and Disclosure of Funding

This work was supported by JST PRESTO (grant number JPMJPR21C6) and a grant from the SECOM Science and Technology Foundation.

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

# A   Bibliographic notes

In this section, we provide additional references intended to complement those in the main body of the paper.

## A.1   Broad overview

The following questions succinctly summarize key aspects of the problem of generalization:[7]

   QP. What properties at training time are reliable indicators of performance at test time?
   QA. How can we efficiently find candidates with such desirable properties?

The easy answer to these questions is, of course, "it depends." There is no fixed procedure that can guarantee arbitrarily good performance on all statistical learning problems, even if restricted to binary classification tasks.[8] A more subtle answer involves characterizing the problems on which abstract learning algorithms such as empirical risk minimization (ERM) yield tight bounds on tractable criteria of interest (e.g., the expected loss).[9] Even more difficult is refining our understanding of the learning problems on which concrete algorithms used in practice can be reliably expected to perform well.[10]

For conceptual grounding, we make use of the two questions, QP (the "property" question) and QA (the "algorithm" question), particularly within the context of non-linear models such as neural networks. Broadly speaking, in the machine learning literature over the past three decades, most answers to the property question QP come in the form of quantifying some notion of "simplicity," a property of candidate $w$. It goes without saying that the underlying heuristic is that all else equal (at training time), a "complex" candidate seems intuitively less likely to perform well at test time.[11] As for the algorithm question, there are numerous "workhorse" procedures of machine learning that are computationally convenient, have high-quality software available, and tend to generalize very well in practice, providing a partial answer to QA. That said, the design principles underlying these procedures are often only very loosely related to the properties that satisfy QP.[12] With this in mind, a large body of research can be understood as trying to develop new connections between answers to QP and QA, either through *post hoc* analysis using existing concepts, or by introducing new properties and deriving algorithms in a more unified fashion.

## A.2   Notions of model complexity

Model complexity is a concept that has a long history in the context of statistical model selection (Claeskens and Hjort, 2008), with well-established ties to information theory (Kullback, 1968). Assuming the quality of "fit" is measured using negative log-likelihood, the second derivative of this objective function (Hessian matrix in multi-dimensional case) is known as the Fisher information (matrix).[13] In the context of neural networks, it is common to use their outputs to model probabilities (Denker and LeCun, 1990), and thus at least conceptually, much of the existing statistical methodology can be inherited. For early work in the context of backprop-driven neural networks, MacKay (1992) looks at designing objective criteria for comparing and choosing between models (including norm regularization parameters). MacKay introduces a form of Bayesian "evidence" for candidate models using a Gaussian approximation that requires evaluating the (inverse) Hessian of the base objective function. More generally, the Hessian describes the curvature of the objective function, and is closely related to geometric notions of "flat regions" on the surface induced by the objective function (Goodfellow et al., 2014; Li et al., 2018).

---

[7]These questions are inspired by the concise and lucid problem setting of Jia and Su (2020).

[8]A lucid explanation is given by Devroye et al. (1996, Ch. 1).

[9]Even this refined problem is far from trivial; see Shalev-Shwartz et al. (2010) and Feldman (2016).

[10]Even the critical question of which core optimizer to use does not have a clear-cut answer on standard benchmark datasets (Schmidt et al., 2021). With additional options such dropout, batch normalization, and all manners of data augmentation, it is not surprising that the practitioner often takes a trial-and-error approach.

[11]In a sense this is just human nature, not specific to machine learning (Baker, 2022).

[12]In the context of training machine learning models both big and small, "Goodhart's law" suggests that this gap is to some extent probably a good thing (https://openai.com/research/measuring-goodharts-law).

[13]Since the data is random, so is the Fisher information; some authors call this the *observed* Fisher information, in contrast with the *expected* Fisher information (Efron and Hinkley, 1978).

The notion of model complexity has also played an central role in statistical learning theory. It has long been known that even when the number of parameters far outnumber the number of training samples, a small weight norm can be used to guarantee off-sample generalization for empirical risk minimizers (Bartlett, 1998). Of course, due to the high expressive power of neural network models, even with strong weight regularization it is possible to perfectly fit random labels (Zhang et al., 2017), leading to a gap between the test error (chance level) and training error (zero) that is methodologically unsatisfactory. This has motivated a variety of new approaches to measure off-sample generalization (Jiang et al., 2020), as well as to quantify model complexity, such as the degree to which a model can be meaningfully compressed (Arora et al., 2018).

The notion of "flat minima" is seen in the early work of Hochreiter and Schmidhuber (1994, 1997), which considers both how to measure sharpness, and heuristics for actually finding candidates in flat regions. The basic underlying notion is that of measuring "volume," namely the idea that a "flat" point is one from which we need to go far in most (if not all) directions for the objective function to increase a certain fixed amount. See more recent work by Wu et al. (2017) for related notions of volume in this context. These notions of sharpness are intimately related to properties of the Hessian matrix of the underlying objective function, even when the loss is not based on negative log-likelihood, and an active line of research is centered around the eigenvalue distribution of this Hessian. See for example Chaudhari et al. (2017) and Karakida et al. (2019) for representative work. For sufficiently "regular" models, the determinant of the Fisher information matrix plays a central role in the complexity term used to implement the minimum description length (MDL) principle (Grünwald, 2007); see also early work from Hinton and van Camp (1993) and more recent work by Jia and Su (2020) in the context of neural networks. More generally, however, many neural networks do not satisfy these regularity conditions, and new technical innovations based on the Fisher information have been explored to bridge this gap in recent years (Sun and Nielsen, 2021).

## A.3   Algorithms that generalize well

The empirical effectiveness of deep learning goes well beyond what we would expect based purely on learning theoretical insights (Sejnowski, 2020). This success is driven by a handful of workhorse stochastic gradient-based solvers (Schmidt et al., 2021), often coupled with explicit norm-based regularization and a number of techniques used to stabilize learning and effectively constrain the model candidate which is selected by the learning algorithm.[14] A rich line of research has developed over the past decade looking at why a certain algorithmic "recipe" tends to generalize well. The tendency for stochastic gradient descent to "escape" from regions near undesirable critical points is one key theme; see Xie et al. (2020) for example. For influential work on relating sharpness, mini-batch size, and (weight) norms to off-sample generalization, see Keskar et al. (2017) and Neyshabur et al. (2017). In both papers, the notion of measuring sharpness by a worst-case perturbation appears, and this is pursued even further by Foret et al. (2021) in the well-known sharpness-aware minimization (SAM) algorithm, and extensions due to Kwon et al. (2021) and Zhao et al. (2022). These algorithms, as well as the Fisher information-based procedure of Jia and Su (2020), all involve a forward-difference implementation of explicit gradient regularization (using squared Euclidean norm), and recent work from Karakida et al. (2023) compares this approach with that of direct back-propagation approach. Barrett and Dherin (2021) look at both *implicit* and *explicit* gradient regularization. The implicit side contrasts the path of "continuous" gradient-based updates with the "discrete" updates made in practice (continuous/discrete with respect to *time*), saying that the discrete updates, even when computed based on an unregularized objective function, tend to move closer to the (continuous) path of a regularized objective, where regularization is in terms of the (squared) gradient norms. Inspired by this finding, they also consider explicit GR in the same way; see also Smith et al. (2021).

---

[14]These include early stopping, modifying mini-batch size, dropout, batch normalization, stochastic depth, data augmentation, and "mixup" (mixed sample augmentations) among others. See also Montavon et al. (2012) for techniques that were established in the decades before the current wave of deep learning.

## B Technical appendix

### B.1 More details on Flooding and sharpness

When the empirical risk goes below the threshold $\theta$, the Flooding update (4) attempts to push it back up above $\theta$. Consider the case in which this occurs in a single step, i.e., the situation in which at some step $t$, the pair of sequential iterates $(w_t, w_{t+1})$ satisfy the following:

$$\mathsf{R}_n(w_t) < \theta \text{ and } \mathsf{R}_n(w_{t+1}) > \theta. \tag{14}$$

When condition (14) holds, some basic algebra immediately shows us that running two iterations of the Flooding update (4) yields the equality

$$w_{t+2} = w_t - \alpha^2 \left( \frac{\nabla \mathsf{R}_n(w_t + \alpha \nabla \mathsf{R}_n(w_t)) - \nabla \mathsf{R}_n(w_t)}{\alpha} \right), \tag{15}$$

telling us that the result is equivalent to running one iteration of FD descent with step size $\alpha^2$ on the GR penalty $\|\nabla \mathsf{R}_n(\cdot)\|^2$ at $w_t$, using the forward FD approximation described earlier in §2.2. To the best of our knowledge, this link was first highlighted by Karakida et al. (2023, §5.1). In a sense, this is a natural complement to the strategy employed in (6); instead of tackling the GR objective $\widetilde{\mathsf{R}}_n(w; \lambda)$ in (5) directly, the Flooding algorithm can iterate back and forth between optimizing the empirical risk and the squared gradient norm. The GR effect is thus constrained to regions in $\mathcal{W}$ with $\theta$-small empirical risk, but all updates outside this region enjoy the same per-step computational complexity as vanilla GD.

### B.2 Non-smooth loss setting

All of the analysis in §3.2 relies heavily on smoothness of the underlying loss function. Here we consider the case in which the loss itself may not even be differentiable. All we ask is that the losses be $L$-Lipschitz on $\mathcal{W}$ in expectation, i.e., $\mathbf{E}_\mu |\ell(w_1; \mathsf{Z}) - \ell(w_2; \mathsf{Z})| \leq L\|w_1 - w_2\|$ for all $w_1, w_2 \in \mathcal{W}$. As an explicit objective function, we start with $\mathsf{S}_\mu$ as given in (10), but with the understanding that $\rho$ can actually be any 1-Lipschitz function, capturing the two special cases of interest, namely $\rho(x) = \sqrt{x^2 + 1} - 1$ for SoftAD and $\rho(x) = |x|$ for Flooding. Shifting our focus to function values (rather than gradients), we will also need to assume a second moment bound

$$\mathbf{E}_\mu \left( \theta + \rho(\ell(w; \mathsf{Z}) - \theta) \right)^2 \leq V < \infty \tag{16}$$

that holds over $w \in \mathcal{W}$. With these basic assumptions in place, stationarity guarantees are available via a smooth approximation of $\mathsf{S}_\mu$.

**Proposition 6** (Stationarity, non-smooth case). *Choosing an initial value $w_1 \in \mathcal{W}$, run the algorithm described in Proposition 3, but re-defining the core gradients used for updating as*

$$\mathsf{G}_t(w) := \frac{d}{r} \left( \theta + \rho(\ell(w + r\mathsf{U}_t; \mathsf{Z}_t) - \theta) \right) \mathsf{U}_t$$

*where $(\mathsf{U}_1, \mathsf{U}_2, \ldots)$ is a sequence of independent vectors sampled uniformly at random from the unit sphere, and $r > 0$ sets the smoothing radius. In addition, the norm threshold is set as $\gamma = \sqrt{dL/((1-b)r)}$ (with $b$ unchanged). Stationarity of the resulting sequence $(w_1, w_2, \ldots)$, assumed to be in $\mathcal{W}$, can be controlled with probability $1 - \delta$ as*

$$\frac{1}{T} \sum_{t=1}^{T} \|\nabla \bar{\mathsf{S}}_\mu(w_t; r)\| \leq \frac{1}{T^{1/4}} \left( \bar{\mathsf{R}}_\mu(w_1) - \bar{\mathsf{R}}_\mu(w_{T+1}) + \frac{3dL}{2r} + \frac{2d}{r}\sqrt{V}\left(1 + C_\delta\right) \right)$$

*where $\bar{\mathsf{S}}_\mu(w; r) := \mathbf{E}[\mathsf{S}_\mu(w + r\mathsf{V})]$ is the $r$-smoothed approximation of the objective $\mathsf{S}_\mu$, with $\mathsf{V}$ distributed uniformly over the unit ball. Probability is taken over the random draw of $(\mathsf{Z}_1, \mathsf{Z}_2, \ldots)$ and $(\mathsf{U}_1, \mathsf{U}_2, \ldots)$, and the confidence factor $C_\delta$ matches that given in Proposition 3.*

*Remark* 7 (Stationarity in the original objective). When the original objective $\mathsf{S}_\mu(\cdot)$ is sufficiently well-behaved, e.g., differentiable almost everywhere and Lipschitz, then stationarity guarantees in terms of the $r$-smoothed objective $\bar{\mathsf{S}}_\mu(\cdot; r)$ can be easily translated into analogous guarantees for $\widetilde{\mathsf{R}}_\mu(\cdot)$. In addition, recent work by Cutkosky et al. (2023) shows how a modified algorithmic approach can be used to achieve faster rates under such congenial (but still non-convex and non-smooth) conditions.

## B.3 Additional proofs

*Proof of Proposition 3.* The machinery of Cutkosky and Mehta (2021, Thm. 2) gives us the ability to control the stationarity of sequences generated using the described procedure (norm-clipping, momentum, normalization), just assuming the "raw" stochastic gradients (here, $\mathsf{G}_t$) are unbiased estimators of a smooth function. As such, we just need to ensure the assumptions underlying their Theorem 2 (henceforth, *CHT2*) are met; the key points have already been described in the main text, so we just fill in the details here. The "unbiased estimator" property we refer to means that we want

$$\mathbf{E}_\mu \, \mathsf{G}_t(w) = \nabla \mathsf{S}_\mu(w) \tag{17}$$

to hold for all $w \in \mathcal{W}$. Fortunately, this holds under very weak assumptions; the running assumption that $L_{\text{AD}} < \infty$ is more than sufficient.[15] In addition, finite $L_{\text{AD}}$ also implies that the objective (10) is smooth in the sense that

$$\|\nabla \mathsf{S}_\mu(w_1) - \nabla \mathsf{S}_\mu(w_2)\| \leq L_{\text{AD}} \|w_1 - w_2\| \tag{18}$$

for any $w_1, w_2 \in \mathcal{W}$; this is proved in §B.5. In addition, uniform second moment bounds naturally imply pointwise bounds, so we have

$$\mathbf{E}_\mu \|\nabla \ell(w; \mathsf{Z})\|^2 \leq \mathbf{E}_\mu \left[ \sup_{w \in \mathcal{W}} \|\nabla \ell(w; \mathsf{Z})\|^2 \right] \leq L_{\text{AD}} - L_\ell \tag{19}$$

for each $w \in \mathcal{W}$. Taken with the construction of sequence $(w_1, w_2, \ldots)$ in the hypothesis, the properties (17)–(19) ensure all the basic requirements of CHT2 are met (with their "$\mathfrak{p}$" at 2). Noting that we assume $\mathcal{W} \subseteq \mathbb{R}^d$ using the standard norm and inner product on Euclidean space, the Banach space generality in CHT2 is not needed (their "$C$" and "$p$" can be fixed to 1 and 2 respectively). For reference, the complete upper bound implied by CHT2 is

$$\frac{\mathsf{S}_\mu(w_1) - \mathsf{S}_\mu(w_{T+1})}{T\alpha} + \frac{\alpha L_{\text{AD}}}{2} + \frac{2b\sqrt{L_{\text{AD}} - L_\ell}}{(1-b)T} + \frac{2b\alpha L_{\text{AD}}}{(1-b)} + 2\sqrt{(1-b)(L_{\text{AD}} - L_\ell)} C_\delta \tag{20}$$

where for readability the coefficient in the right-most summand is defined by

$$C_\delta := 10 \log(3T/\delta) + 4\sqrt{\log(3T/\delta)} + 1.$$

We have simplified all the terms in CHT2 involving $\max\{1, \log(3T/\delta)\}$, since as long as $T > 0$ and $0 < \delta < 1$, we trivially have $3T/\mathrm{e} \geq 1 > \delta$ and thus $\log(3T/\delta) \geq 1$. Furthermore, their free parameters "$b$" (different from our $b$) and "$s$" are both set to 1, without loss of generality. Plugging in our settings of $\alpha$ and $b$ to the bound in (20) and bounding $(1 - 1/\sqrt{T}) \leq 1$ for readability yields the desired upper bound. $\qquad\square$

*Proof of Proposition 4.* Here we leverage the projected sub-gradient analysis done by Davis and Drusvyatskiy (2019), in particular their Theorem 3 (henceforth, *DDT3*). The core of their argument relies upon a weak convexity property held by a rather large class of composite functions, namely compositions of the form $f = h \circ g$, where $g$ is smooth and $h$ is both convex and Lipschitz. Considering the non-smooth objective function

$$w \mapsto \theta + \mathbf{E}_\mu |\ell(w; \mathsf{Z}) - \theta|, \tag{21}$$

it can be taken as a compound function by writing $\mathbf{E}_\mu \, f(w; \mathsf{Z})$ with $f(w; z) := h(g(w; z))$, where $g(w; z) := \ell(w; z)$ and $h(x) := \theta + |x - \theta|$. Fixing $z \in \mathcal{Z}$ for now, clearly $h$ is 1-Lipschitz and convex. By assumption, we have that $g(\cdot; z)$ is $L_\ell^*$-smooth and locally Lipschitz. Then, using standard arguments, it is straightforward to show that $f(\cdot; z)$ is $L_\ell^*$-weakly convex.[16] Since the weak convexity parameter $L_\ell^*$ does not depend on the arbitrary choice of $z$, it follows that the function (21) is $L_\ell^*$-weakly convex. In the setting of DDT3, their "$f(\cdot)$" is $\theta + \mathbf{E}_\mu |\ell(w \cdot; \mathsf{Z}) - \theta|$, and their "$\mathcal{X}$" is $\mathcal{W}$ here. The bound on the expected squared stochastic gradient norms (their "$L^2$") is our $L_{\text{AD}} - L_\ell$ just as in the proof of Proposition 3. Finally, the key "unbiased estimator" property in this case deals with sub-differentials, namely we require that

$$\mathbf{E}_\mu \, \mathsf{G}_t(w) \in \partial \, \mathbf{E}_\mu |\ell(w \cdot; \mathsf{Z}) - \theta| \tag{22}$$

---

[15] For a more general result, see Holland (2023, Lem. 2) for example.

[16] See for example Drusvyatskiy and Paquette (2019, Lem. 4.2) and Holland (2022, Prop. 8).

for all $w \in \mathcal{W}$. Fortunately this basic property holds under very weak assumptions that are trivially satisfied when $L_{\text{AD}}$ is finite.[17] With these facts in place, we simple apply DDT3, in particular their inequality (3.5), with their "$\rho$" corresponding to our $L_\ell^*$ here, and their "$\varphi_\lambda$" corresponding to our (13), with "$\lambda$" as our $\beta$. The desired bound follows by applying their result to the specified procedure over $T - 1$ updates (instead of their $T$ updates). □

*Proof of Proposition 6.* To begin, under the assumptions given, the objective $S_\mu$ clearly inherits the Lipschitz property that the losses have in expectation; since $\rho$ is 1-Lipschitz, we have

$$|S_\mu(w_1) - S_\mu(w_2)| \leq \mathbf{E}_\mu |\ell(w_1; Z) - \ell(w_2; Z)| \leq L\|w_1 - w_2\|. \tag{23}$$

We proceed by using a standard technique for function smoothing.[18] If we let $V$ be a random vector distributed over the unit ball $\{x \in \mathbb{R}^d : \|x\| \leq 1\}$, then regardless of whether $S_\mu(\cdot)$ is differentiable or not, one can obtain a smooth approximation by averaging over random $r$-length perturbations, namely

$$\bar{S}_\mu(w; r) := \mathbf{E}\left[S_\mu(w + rV)\right]. \tag{24}$$

A critical property of the function given in (24) is that it is differentiable and its gradient can be represented explicitly in terms of the function it is trying to smooth, namely we have

$$\nabla \bar{S}_\mu(w; r) = \frac{d}{r} \mathbf{E}\left[S_\mu(w + rU)U\right] = \frac{d}{r} \mathbf{E}\left[(\theta + \mathbf{E}_\mu \, \rho(\ell(w + rU) - \theta)) U\right] \tag{25}$$

for any choice of $r > 0$ and $w \in \mathcal{W}$, where $U$ is uniformly distributed on the unit sphere $\{x \in \mathbb{R}^d : \|x\| = 1\}$ (Flaxman et al., 2004, Lem. 1).[19] This means that Lipschitz properties on the original function translate to smoothness properties for the new function. Making this more explicit, using the equality (25), note that for any choice of $w_1, w_2 \in \mathcal{W}$, we have

$$\nabla \bar{S}_\mu(w_1; r) - \nabla \bar{S}_\mu(w_2; r) = \frac{d}{r} \mathbf{E}\left[(S_\mu(w_1 + rU) - S_\mu(w_2 + rU)) U\right].$$

Taking norms and using the Lipschitz property (23) of $S_\mu$, we observe that

$$|\nabla \bar{S}_\mu(w_1; r) - \nabla \bar{S}_\mu(w_2; r)| \leq \frac{d}{r} \mathbf{E}\|U\| |S_\mu(w_1 + rU) - S_\mu(w_2 + rU)|$$

$$\leq \frac{dL}{r}\|w_1 - w_2\|$$

and thus have that the smoothed function $\bar{S}_\mu(\cdot; r)$ is $(dL/r)$-smooth over $\mathcal{W}$. This means the function is analogous to the objective function (10) used in Proposition 3, except with unbiased stochastic gradients taking the form

$$G_t(w) := \frac{d}{r}\left(\theta + \rho(\ell(w + rU_t; Z_t) - \theta)\right) U_t \tag{26}$$

for $t \geq 1$, where each $U_t$ is an independent copy of $U$ from (25). From this point, the remainder of the proof is basically identical to that of Proposition 3; the only remaining changes are the smoothness factor and the second moment bound. For the former, we use $dL/r$ in place of $L_{\text{AD}}$, which also impacts the norm clipping radius $\gamma$. For the latter, since we are assuming $w_t + rU_t \in \mathcal{W}$ for each $t$, and using the bound (16), we have

$$\mathbf{E}\|G_t(w)\|^2 \leq \sup_{w \in \mathcal{W}} \left(\frac{d}{r}\right)^2 \mathbf{E}\|U_t\|^2 \left(\theta + \rho(\ell(w; Z_t) - \theta)\right)^2 \leq \left(\frac{d}{r}\right)^2 V.$$

Plugging in these two remaining modified factors to the bounds obtained in Proposition 3 yields the desired result. □

---

[17]See for example Holland (2022, Prop. 14).

[18]See for example Flaxman et al. (2004); Nesterov and Spokoiny (2017).

[19]Not to be confused with $V$ in (24), which is uniform on the unit *ball*.

## B.4 Gradient of GR objective

With $w = (w_1, \ldots, w_d) \in \mathbb{R}^d$, we will frequently use $\partial_j$ to denote partial derivatives taken with respect to $w_j$, i.e., for a differentiable function $f : \mathbb{R}^d \to \mathbb{R}$, we write

$$\partial_j f(w) := \lim_{|a| \to 0} \frac{f(w_1, \ldots, w_j + a, \ldots, w_d) - f(w)}{a} \tag{27}$$

with analogous definitions for all $j = 1, \ldots, d$. With the above notation in place, note that basic calculus gives us

$$\partial_j \|\nabla \mathsf{R}_n(w)\|^2 = \sum_{k=1}^d \partial_j \left( \partial_k \mathsf{R}_n(w) \right)^2 = 2 \sum_{k=1}^d \left( \partial_k \mathsf{R}_n(w) \right) \left( \partial_j \partial_k \mathsf{R}_n(w) \right).$$

As such, the gradient takes the form

$$\nabla \|\nabla \mathsf{R}_n(w)\|^2 = 2 \nabla^2 \mathsf{R}_n(w) \left( \nabla \mathsf{R}_n(w) \right)$$

where $\nabla^2 \mathsf{R}_n$ denotes the $d \times d$ Hessian matrix of $\mathsf{R}_n$, and $\nabla \mathsf{R}_n(w)$ is taken as a column vector (a $d \times 1$ matrix) for the purpose of this multiplication.

## B.5 Smoothness check

Let us consider a simple, non-stochastic example in one dimension. Letting $f : \mathbb{R} \to \mathbb{R}$ be some differentiable function, we consider how the transformed gradient $\phi(f(x) - \theta)f'(x)$ behaves under $\phi = \operatorname{sign}$ and $\phi = \rho'(x) = x/\sqrt{x^2 + 1}$. As we have seen visually in Figure 1, the soft threshold of SoftAD makes it possible to have Lipschitz gradients, which impacts iterative optimization procedures. For arbitrary values $x_1$ and $x_2$, taking the difference of transformed gradients using arbitrary $\phi$, we can write

$$\phi(f(x_1) - \theta)f'(x_1) - \phi(f(x_2) - \theta)f'(x_2)$$
$$= \left( \phi(f(x_1) - \theta) - \phi(f(x_2) - \theta) \right) f'(x_1) + \phi(f(x_2) - \theta) \left( f'(x_1) - f'(x_2) \right). \tag{28}$$

Note that even if $|x_1 - x_2| < \varepsilon$ for some arbitrarily small $\varepsilon > 0$, if for example the threshold is such that $f(x_1) < \theta < f(x_2)$, then under $\phi = \operatorname{sign}$, the difference multiplying $f'(x_1)$ cannot be arbitrarily small, even if $f$ is Lipschitz. On the other hand, such a property follows easily when $\phi = \rho'$, since $\rho'$ itself is 1-Lipschitz.

Returning to our more general learning setup, let us denote the modified gradients concisely as $g(w; z) := \phi(\ell(w; z) - \theta)\nabla \ell(w; z)$. With random variable $\mathsf{Z} \sim \mu$, taking any two points $w_1, w_2 \in \mathcal{W}$, based on the equality (28), the normed difference of the gradient expectations can be bounded as

$$\|\mathbf{E}_\mu \, g(w_1; \mathsf{Z}) - \mathbf{E}_\mu \, g(w_2; \mathsf{Z})\| \leq B_1 + B_2$$

with $B_1$ and $B_2$ defined as

$$B_1 := \mathbf{E}_\mu \|\nabla \ell(w_1; \mathsf{Z})\| |\phi(\ell(w_1; \mathsf{Z}) - \theta) - \phi(\ell(w_2; \mathsf{Z}) - \theta)|$$
$$B_2 := \mathbf{E}_\mu |\phi(\ell(w_2; \mathsf{Z}) - \theta)| \|\nabla \ell(w_1; \mathsf{Z}) - \nabla \ell(w_2; \mathsf{Z})\|.$$

Bounding each of these terms is trivial when the functions $\ell$ and $\phi$ are smooth enough. First, note that if $\phi$ is $L_\phi$-Lipschitz and $\mathcal{W}$ is a convex subset of $\mathbb{R}^d$, we have

$$B_1 \leq L_\phi \, \mathbf{E}_\mu \|\nabla \ell(w_1; \mathsf{Z})\| |\ell(w_1; \mathsf{Z}) - \ell(w_2; \mathsf{Z})|$$
$$\leq L_\phi \|w_1 - w_2\| \, \mathbf{E}_\mu \|\nabla \ell(w_1; \mathsf{Z})\| \sup_{0 < a < 1} \|\nabla \ell(aw_1 + (1-a)w_2; \mathsf{Z})\|$$
$$\leq L_\phi \|w_1 - w_2\| \, \mathbf{E}_\mu \left[ \sup_{w \in \mathcal{W}} \|\nabla \ell(w; \mathsf{Z})\|^2 \right],$$

noting that the second inequality uses the mean value theorem on differentiable $\ell(\cdot; z)$, applied pointwise in $z \in \mathcal{Z}$, and the last inequality uses convexity of $\mathcal{W}$. This bounds $B_1$. Moving on to $B_2$, note that if $|\phi(x)|$ is bounded by $B_\phi$ and the losses are $L_\ell$-smooth in expectation, we have

$$B_2 \leq B_\phi \, \mathbf{E}_\mu \|\nabla \ell(w_1; \mathsf{Z}) - \nabla \ell(w_2; \mathsf{Z})\|$$
$$\leq B_\phi L_\ell \|w_1 - w_2\|.$$

Taking these new bounds together, we have

$$\|\mathbf{E}_\mu \, g(w_1; \mathsf{Z}) - \mathbf{E}_\mu \, g(w_2; \mathsf{Z})\| \leq \left( L_\phi \, \mathbf{E}_\mu \left[ \sup_{w \in \mathcal{W}} \|\nabla \ell(w; \mathsf{Z})\|^2 \right] + B_\phi L_\ell \right) \|w_1 - w_2\|,$$

namely a Lipschitz property in expectation for the modified gradients.

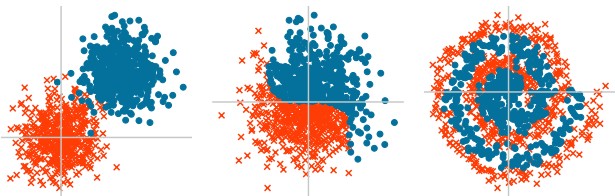

**Figure 7:** Synthetic dataset examples. From left to right: "two Gaussians," "sinusoid," and "spiral."

# C   Empirical appendix

Here we provide additional details and results related to the empirical tests described in §4.

**Software and hardware**   All of the experiments done in this section have been implemented using PyTorch 2, utilizing three machines each using a single-GPU implementation, i.e., there is no parallelization across multiple machines or GPUs. Two units are equipped with an NVIDIA A100 (80GB), and the remaining machine uses an NVIDIA RTX 6000 Ada. We use the MLflow library for storing and retrieving metrics and experiment details. Our coding of SAM follows that of David Samuel (`https://github.com/davda54/sam`), which is the PyTorch implementation acknowledged in the original SAM paper of Foret et al. (2021).

## C.1   Non-linear binary classification on the plane

**Data**   The three types of synthetic data that we generate here differ chiefly in the degree of non-linearity; see Figure 7 for an example. The "two Gaussians" dataset is almost linearly separable, save for some overlap of the two distributions. The "sinusoid" data is separated by a simple curve, easily approximated by a low-order polynomial, but the curves in the "spiral" data are a bit more complicated. Exact implementation details, plus historical references, are given by Ishida et al. (2020, §4.1). For each trial, we generate training and validation data of size 100, and test data of size 20000. All methods see the same data in each trial.

**Model**   For each dataset, we use the same model, namely a simple feedforward neural network with four hidden layers, 500 units per layer, using batch normalization and ReLU activations at each layer.[20]

**Algorithms**   In line with the experiments we are trying to replicate, all methods (ERM, Flooding, SAM, and SoftAD) are driven by the Adam optimizer, using a fixed learning rate of 0.001, with no momentum or weight decay. All methods use the multi-class logistic loss as their base loss (i.e., `nn.CrossEntropyLoss` in PyTorch), and are run for 500 epochs. We use mini-batch size of 50 here, but key trends remain the same for full-batch (of size 100) runs.

**Hyperparameter selection**   ERM has no hyperparameters, but all the other methods have one each. Flooding and SoftAD both have the threshold parameter $\theta$ seen in §2–§3, and SAM has a radius parameter (denoted "$\rho$" in the original paper). For each of these methods, in each trial, we select from a grid of 40 points spaced linearly between 0.01 and 2.0. Selection is based on classification accuracy on validation data.

## C.2   Image classification from scratch

Our second set of experiments utilizes four well-known benchmark datasets for multi-class image classification. Compared to the synthetic experiments done in §C.1, the classification task is more difficult (much larger inputs, variation within classes, more classes), and so we utilize more sophisticated neural network models to tackle the classification task. That said, as the sub-section title indicates, this training is done "from scratch," i.e., no pre-trained models are used.

---

[20]Ishida et al. (2020) say they use a "five-hidden-layer feedforward neural network," but looking at their public code, the number of hidden layers (i.e., number of linear transformations excluding that of the output layer) is actually four.

**Data**   The datasets we use are all standard benchmarks in the machine learning community: CIFAR-10, CIFAR-100, FashionMNIST, and SVHN. All of these datasets are collected using classes defined in the `torchvision.datasets` module, with raw training/test splits left as-is with default settings. As such, across all trials the test set is constant, but in each trial we randomly select 80% of the raw training data to be used for actual training, with the remaining 20% used for validation. We normalize all pixel values in the image data to the unit interval $[0, 1]$; this is done separately for training, validation, and testing data.

**Models**   Unlike the previous sub-section, here we use different models for different data sets. Model choice essentially mirrors that of Ishida et al. (2020, §4.2). For FashionMNIST, we flatten each image into a vector, and use a simple feedforward neural network composed of a single hidden layer with 1000 units, batch normalization, and ReLU activation before the output transformation. For SVHN, we use ResNet-18 as implemented in `torchvision.models`, without any pre-trained weights. Finally, for both CIFAR-10 and CIFAR-100, we use ResNet-34 (again in `torchvision.models`) without pre-training. Both of the ResNet models used do not flatten the images, but rather take each RGB image as-is.

**Algorithms**   Just as in §C.1, we are testing ERM, Flooding, SAM, and SoftAD. Again we use the cross entropy loss, and run for 500 epochs. However, instead of Adam as the base optimizer, here we use vanilla SGD with a fixed step size of $0.1$, and momentum parameter of $0.9$. For all datasets, we use a mini-batch size of 200. All these settings match the experimental setup of Ishida et al. (2020, §4.2).[21]

**Hyperparameter selection**   Once again we select hyperparameters for Flooding, SoftAD, and SAM from a grid of candidate values, such that the classification accuracy on validation data is maximized. Unlike §C.1 however, here we use different grids for each method. For Flooding, we follow the setup of the original paper, choosing from ten values: $\{0.01, 0.02, \ldots, 0.1\}$. For SAM, once again we follow the original paper (their §3.1), which for analogous tests utilized the set $\{0.01, 0.02, 0.05, 0.1, 0.2, 0.5\}$. Finally, for SoftAD we match the set size used by Flooding (i.e., ten) by taking the union of $\{0.15, 0.25, 0.35, 0.75\}$ and the set used by SAM.

---

[21]Batch size is not given in the original Flooding paper, but the size of 200 was confirmed by means of a private communication with the authors.

**Comparison with iFlood**  As mentioned in Remark 1, during the review phase of this work, the iFlood method of Xie et al. (2022) was brought to our attention, and we have run additional tests analogous to those described in the preceding paragraphs, but this time comparing ERM, iFlood, and SoftAD. The results are given below in Table 3, Figures 8–10, and Table 4, in that order.

| | CIFAR-10 | CIFAR-100 | Fashion | SVHN |
|---|---|---|---|---|
| ERM | 3.252 | 7.695 | 0.806 | 0.726 |
| iFlood | 1.468 | **2.558** | **0.273** | 0.404 |
| SoftAD | **1.072** | 2.696 | 0.463 | **0.397** |

**Table 3:** Generalization gap (test - training) for trial-averaged cross entropy loss after final epoch.

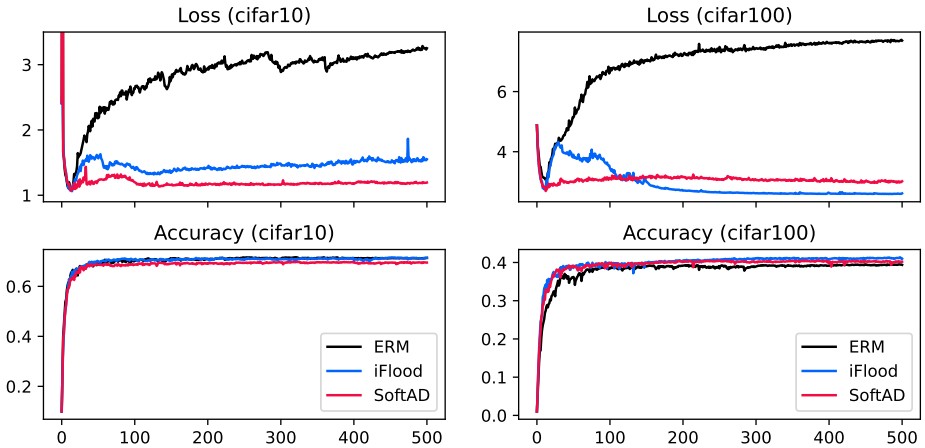

**Figure 8:** Trajectories over epochs for average test loss (top row) and test accuracy (bottom row). Horizontal axis is epoch number. Columns are associated with the CIFAR-10 and CIFAR-100 datasets (left to right).

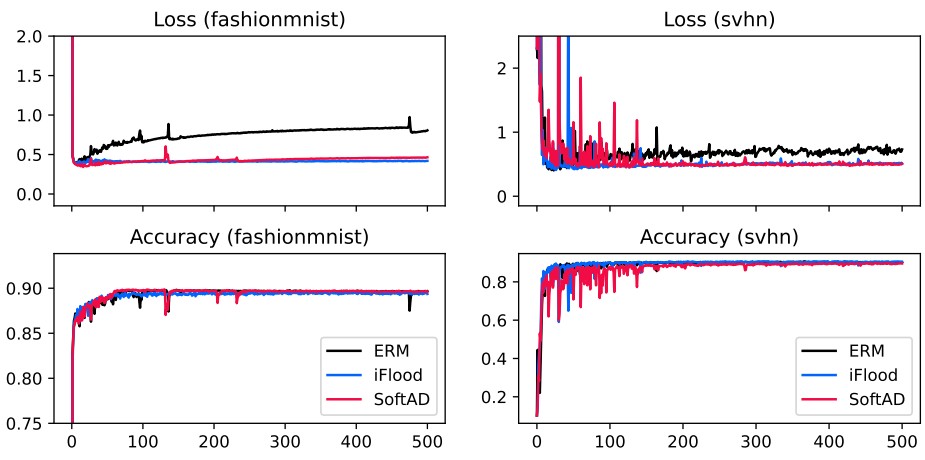

**Figure 9:** Analogous to Figure 8, but with FashionMNIST and SVHN datasets.

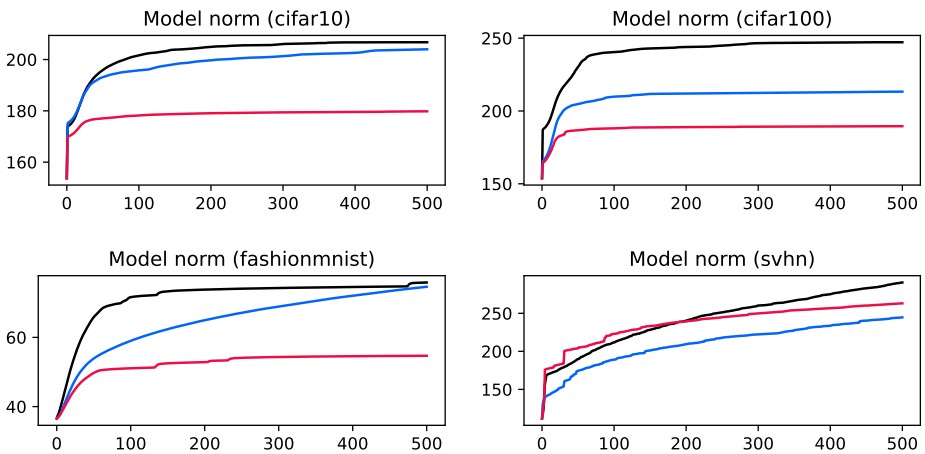

**Figure 10:** Model norm trajectories over epochs for each dataset in Figures 8–9.

| | CIFAR-10 | CIFAR-100 | Fashion | SVHN |
|---|---|---|---|---|
| iFlood | 0.04 (0.05) | 0.06 (0.04) | 0.15 (0.10) | 0.10 (0.05) |
| SoftAD | 0.12 (0.06) | 0.34 (0.28) | 0.01 (0) | 0.12 (0.12) |

**Table 4:** Hyperparameters selected by validation for each method (averaged over trials). Flooding and SoftAD have threshold $\theta$; SAM has radius parameter. Standard deviation (over trials) is given in small-text parentheses.

## C.3 Linear binary classification

In Figure 11, we compare ERM with Flood and SoftAD run with a *common* threshold level of $\theta = 0.25$, using the "two Gaussians" and "sinusoid" data described in §C.1, and a simple linear model, i.e., a feed-forward neural network with no hidden layers. Training and test sizes match those described in §C.1. Even with a very simple linear model, it is clear that SoftAD can be used to achieve competitive accuracy at much larger loss levels. Note that in the case of "sinusoid," the average loss does not reach the threshold $\theta$, and thus Flooding is identical to ERM. These basic trends hold over a range of thresholds $\theta$ and re-scaling parameters $\sigma$ (i.e., using $\phi((x - \theta)/\sigma)$ with $\sigma \neq 1$). These trends are captured by the heatmaps given in Figure 12, where for each setting of $\theta$ (for SoftAD and Flooding) and $\sigma$ (for SoftAD only), we generate a fresh dataset. Clearly taking the threshold level far too high leads to arbitrarily bad performance, but below a certain level, similar performance is observed over a wide range of values. It is interesting to note how while test loss changes in a rather predictable continuous fashion as a function of $\theta$, the test *accuracy* drops in a much sharper manner when $\theta$ is set too high in the case of SoftAD, whereas this drop is smoother in the case of Flooding. That said, these trends are only within the confines of this very simple linear model example using full batch, and tend to change (even with the same model) as we modify the mini-batch size.

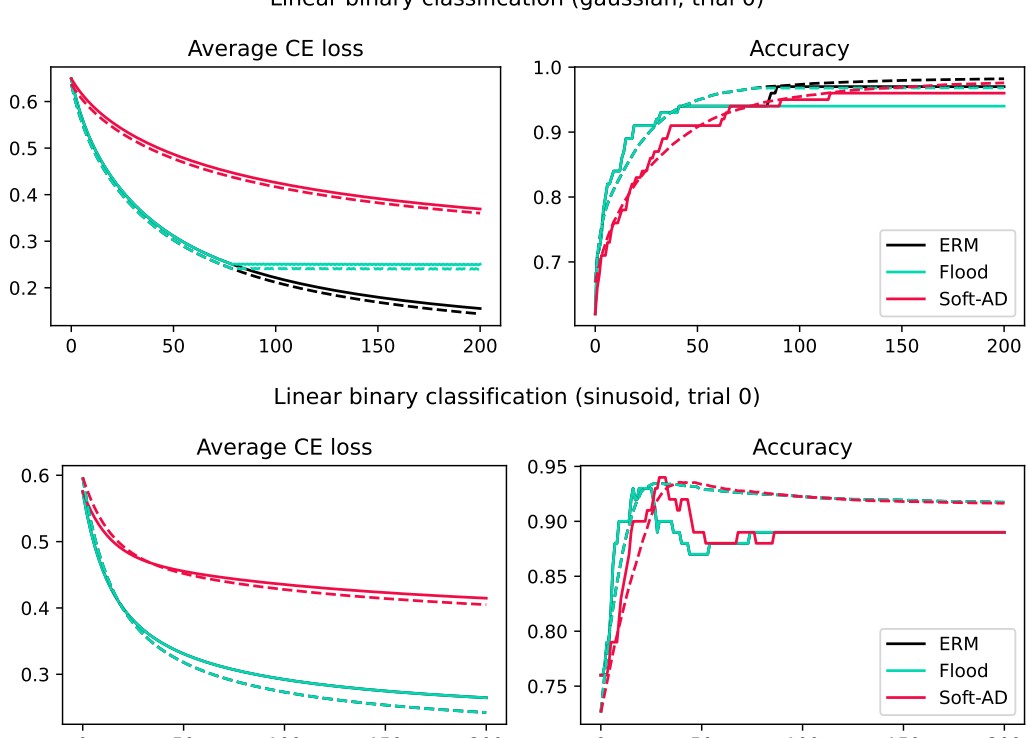

**Figure 11:** Average cross entropy loss and accuracy over epochs (full batch) for each method.

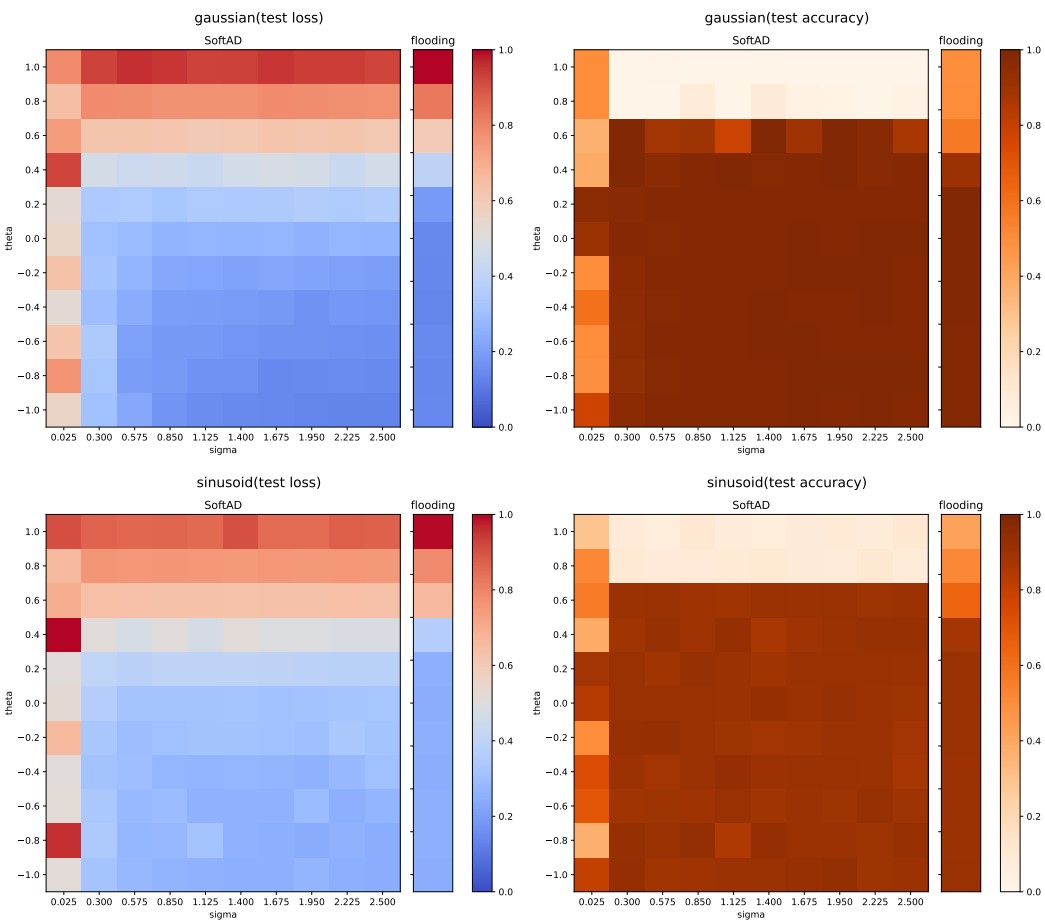

**Figure 12:** Test loss and accuracy heatmaps for Flooding and SoftAD, depending on threshold level (denoted "theta") and scaling parameter (denoted "sigma").

