# OpenReview forum: "Soft ascent-descent as a stable and flexible alternative to flooding"
_NeurIPS.cc/2024/Conference — NeurIPS 2024 poster_

### Official Review · Reviewer_bEZy · 2024-06-27

**Soundness:** 3
**Presentation:** 3
**Contribution:** 2
**Rating:** 6
**Confidence:** 3

**Summary:**

Flooding method is a previous method which aims at improving the generalization performance by making the average loss equal to a given threshold. This paper makes two changes of the 'flooding' method: 1. Rather than making the average loss equal to a given threshold, they make the point-wise loss equal to a given threshold. 2. To make them equal, previous work uses the absolute function to measure the difference, this work uses a softer function. Experiments are conducted to support their method.

**Strengths:**

This paper is easy to follow. The organization of this paper is clear.
The proposed link between flooding and sharpness, as shown in 2.2, is interesting and insightful. It explains why flooding method works from the flatness perspective.

**Weaknesses:**

Although the generalization gap of the proposed method is smaller (shown in Table 1), the accuracy is not larger than SAM (shown in Figure 4). What are the advantages of the proposed method compared with SAM?

**Questions:**

None

---

> ### Author Rebuttal · Authors · 2024-08-02
>
> Thanks for taking the time to read our paper, write a review, and for the positive outlook on our work. Below, we respond to the question you raised.
>
> > Although the generalization gap of the proposed method is smaller (shown in Table 1), the accuracy is not larger than SAM (shown in Figure 4). What are the advantages of the proposed method compared with SAM?
>
> If accuracy at test time is the only metric of interest, and computational cost is not a concern, then SAM is perfectly fine. On the other hand, our tests highlight how with a slight drop in accuracy, SoftAD can achieve big gains in terms of other metrics such as the test-time average surrogate loss, generalization gap, and model norm size. In addition, the implementation of SoftAD is very simple, and computational cost is half of that of SAM (one gradient computed by backprop, instead of two). As such, when multiple performance metrics are of interest, and we are interested in achieving good *tradeoffs* between these metrics, then we believe the main results of this paper are encouraging initial evidence for the utility of SoftAD.
>
> Thanks again for the feedback. If you have any additional questions, we are happy to respond at any time.

---

> > ### Comment · Reviewer_bEZy · 2024-08-08
> >
> > Thanks for your reply! It addresses my questions. But I think the contributions are limited. Therefore, I will remain my score.

---

> > > ### Author Response · Authors · 2024-08-09
> > > **Re: Official Comment by Reviewer bEZy**
> > >
> > > Thanks very much for the follow-up, and for sticking with your positive score. Best regards.

---

### Official Review · Reviewer_yxTo · 2024-06-28

**Soundness:** 3
**Presentation:** 4
**Contribution:** 3
**Rating:** 7
**Confidence:** 4

**Summary:**

The flooding method sets a threshold for the average surrogate loss within a mini-batch during training: if above it do gradient descent as usual but if below it switch to gradient ascent. This paper updates the flooding method in 2 ways: 1) it invert the order of the sign function and the aggregation making it a pointwise method rather than an average method and 2) it replaces the sign function with a continuous, bounded, monotonic function. It provides numerical results to demonstrate the behavior of the new proposed method in Section 3.1 and show theoretical results to compare the convergence properties of the new method with flooding in Section 3.2. Finally, experiments show that the proposed method have similar accuracy compared with flooding/SAM while having much lower generalization gap and smaller model norms (despite not using any explicit shrinkage methods).

**Strengths:**

- The soft truncation is a nice idea to improve the flooding method. The behavior shown in Fig. 1 & 2 is intuitive.
- Stationarity guarantees are provided in the theoretical studies and the intuition is provided in Remark 4.
- Experiments show softAD has the same level of test accuracy (SAM may be slightly better but at the double the gradient cost) while achieving better level of surrogate test loss. Furthermore, the generalization gap is much smaller and model norms are usually smaller than other baselines. The flood level comparisons are interesting: the softAD allows for a larger flood level.
- The discussions about links to sharpness and OCE-like criteria are interesting and informative.
- The story, organization, and presentation is clear and easy to read. There are a lot of informative information in the appendix as well.

**Weaknesses:**

- A similar pointwise idea was proposed in "iFlood: A Stable and Effective Regularizer" (ICLR 2022). It would be helpful to have some discussions about iFlood and explain if there are any differences in the pointwise idea.
- The experiments are informative, clearly showing the benefits of the approach. However, given that the pointwise idea is already proposed, as an ablation study, it would be important to further check which parts are more effective: the pointwise component or the soft truncation component. If the paper can show that the soft truncation is important, that would make the paper's contribution clear.

**Questions:**

(I wrote my main suggestions in the "Weaknesses section". Here I provide some other comments/questions.)

- Having a low generalization gap and better surrogate loss seems to be one strong benefit of the approach. I wonder if the paper can check calibration (such as ECE) which is often correlated with lower test losses.
- It is interesting that softAD allows a higher theta threshold. Is it possible to provide some discussions on why the proposed method allows a higher theta?

**Limitations:**

I believe there is no immediate negative societal impact of this work. The paper discuss that hyper-parameter selection remains and provide future directions to remove this aspect.

---

> ### Author Rebuttal · Authors · 2024-08-02
>
> We thank the reviewer for their time and constructive feedback provided in the review. Below, we respond to the points raised.
>
> __Regarding iFlood (ICLR 2022):__
>
> Thank you very much for this reference. We were completely unaware of this work, but it is a very nice paper, and as the reviewer points out, it is essentially a midpoint between our proposed SoftAD and the original flooding procedure, using pointwise aggregation, but without the smoothing effect that aids in truncating outliers and down-weighting borderline points.
>
> We commit to adding this reference, with discussion, to our paper, and we also commit to adding tests in our setup using iFlood to investigate the effective impact of smoothing on the performance metrics studied in our experiments. The experiments will not conclude by August 6th, but we fully commit to adding such results for a camera-ready version.
>
> ---
>
> __Other comments:__
>
> > Having a low generalization gap and better surrogate loss seems to be one strong benefit of the approach. I wonder if the paper can check calibration (such as ECE) which is often correlated with lower test losses.
>
> Thanks for this very natural question, this is a point we are indeed quite interested in, as mentioned above in the response to Reviewer JmzP. We do have some nascent test results in this direction already, and while an in-depth study is going to be the topic of a future paper, we can certainly add some early results to this paper to complement the insights obtained in terms of test loss performance.
>
> > It is interesting that softAD allows a higher theta threshold. Is it possible to provide some discussions on why the proposed method allows a higher theta?
>
> Yes, we can definitely discuss this in the paper in more detail, plus provide some supplementary figures. In essentially all cases, when we select $\\theta$ based on validation set accuracy, we find that SoftAD ends up with a larger value of $\\theta$ than Flooding. Intuitively, we interpret this as resulting from the fact that SoftAD is qualitatively asking for something quite different from Flooding, namely asking for "good concentration around $\\theta$", instead of just "average loss value close to $\\theta$." After a sufficient amount of training, losses often become quite asymmetrically distributed, and it can be infeasible to achieve high concentration at a threshold $\\theta$ too close to the minimal possible value. This property is most directly due to the point-wise form of the SoftAD objective, and we believe the iFlood method would also show similar properties when compared with Flooding in our experimental setup; we will add such results.
>
> Thanks again for the constructive feedback and overall positive review. If there are any additional points you would like to discuss, please feel free to let us know.

---

> > ### Comment · Reviewer_yxTo · 2024-08-08
> >
> > Thank you for answering my questions.
> >
> > I think the remaining limitation is the lack of emphasis of the softening idea over the pointwise idea.
> >
> > It seems the authors are currently working on such experiments. If the authors have early results for just 1 dataset/setup, that will be helpful for reviewers.
> >
> > Additionally, would it be straightforward to modify the theoretical results in Section 3.2 to emphasize the softening effect by comparing it with iFlood? If yes, this can also clarify the contributions of this paper.

---

> > > ### Author Response · Authors · 2024-08-08
> > > **Re: Official Comment by Reviewer yxTo**
> > >
> > > Thanks very much for the follow-up.
> > >
> > > > If the authors have early results for just 1 dataset/setup, that will be helpful for reviewers.
> > >
> > > Certainly. We're out of the office for the day, but should have initial results organized to share within 24 hours.
> > >
> > > > Additionally, would it be straightforward to modify the theoretical results in Section 3.2 to emphasize the softening effect by comparing it with iFlood?
> > >
> > > This is a great point to raise, and we should have been more explicit about it in our initial rebuttal. The short answer is yes, and in fact, the currently stated results are essentially guarantees for iFlood, though we were unaware of the iFlood paper when writing the original manuscript. To be a bit more precise, after equation (12), we mention that the mini-batch gradients used by the original Flooding procedure do not provide unbiased estimates of (sub-)gradients of the Flooding objective, but they are unbiased estimators for a different objective, namely the upper bound we write as "$\\theta + \\mathbf{E}\_{\\mu}\\vert \\ell(w;\\mathsf{Z}) - \\theta \vert$." This is precisely the population/test objective underlying iFlood. This is also noted by Xie et al., who briefly mention how the point-wise change yields unbiased estimates (end of paragraph 2, sec. 3.1). As such, compared with SoftAD, all the theoretical limitations we highlight in our section 3.2 for Flooding also hold as-is for iFlood. This relation will be made explicit in the revised manuscript.

---

> > > > ### Author Response · Authors · 2024-08-09
> > > > **Re: Official Comment by Reviewer yxTo**
> > > >
> > > > Apologies for the delay, this is a follow-up regarding the initial experimental results using iFlood.
> > > >
> > > > We ran all methods in precisely the setup described in the paper, but this time included iFlood, which is essentially implemented by replacing the smooth function $\\rho(x) = \\sqrt{x^{2}+1} - 1$ used in SoftAD with the absolute value function $\\rho(x) = \\vert x \\vert$. We give iFlood the exact same hyperparameter grid as SoftAD, and choose based on validation accuracy as with the original tests.
> > > >
> > > > The tests for FashionMNIST have already finished, and so we share those here. We cannot attach figures it seems, and to keep anonymity we will just post numerical values here. In the table below, we post values (averaged over trials) for each of the key quantities shown in Figures 4 and 5 in our paper, namely test (average) loss, test accuracy, and model $\\ell\_{2}$ norm, all after the final epoch is finished.
> > > >
> > > > |  | ERM | Flooding | SAM | SoftAD | iFlood |
> > > > | ----------- | ----------- | ----------- | ----------- | ----------- | ----------- |
> > > > | Test loss | $0.80$ | $0.44$ | $0.61$ | $0.45$ | $0.42$ |
> > > > | Test acc | $0.90$ | $0.89$ | $0.90$ | $0.90$ | $0.89$ |
> > > > | $\\ell\_{2}$ norm | $75$ | $84$ | $129$ | $54$ | $74$ |
> > > >
> > > > Essentially, for the small/easy FashionMNIST task, we see that losses and accuracies are quite similar, but a big gap in the model norm appears between methods. Regarding how the trajectory of iFlood is over epochs in each of these quantities, it is very similar to SoftAD in the case of loss and accuracy (as seen in Figure 4 in the paper), and doesn't have the dramatic "double descent" that is only present in Flooding. On the other hand, for the 500 epoch tests we've run, iFlood leads to a model norm growing monotonically (like SAM in our Figure 5). While its growth is slower than SAM, the norm growth does not slow down quickly like in SoftAD. It is natural to infer that as learning progresses, the effect of our smoothing function (downweighting points that are borderline, i.e., essentially good enough) is appearing in this rather stark difference in norm trajectories.
> > > >
> > > > Regarding which value of $\\theta$ is selected by validation data, iFlood has mean $0.15$ with standard deviation $0.1$, clearly more variance than we see in Flooding (mean $0.01$, SD $0.0$) and SoftAD (mean $0.03$, SD $0.04$) for the same dataset, though this might settle down with more trials.
> > > >
> > > > Taken together, it seems like the pointwise design (in both SoftAD and iFlood) leads to a faster convergence than vanilla Flooding, but the smoothness in SoftAD is what induces the strong regularization effect on the model norm, especially in the later stages of the learning process. Thanks again for the feedback.

---

> > > > > ### Comment · Reviewer_yxTo · 2024-08-09
> > > > >
> > > > > Thank you for the update. The contribution of the paper is more clear now. Since I don't have any more questions/concerns, I adjusted my score.

---

### Official Review · Reviewer_JmzP · 2024-07-18

**Soundness:** 2
**Presentation:** 3
**Contribution:** 2
**Rating:** 4
**Confidence:** 3

**Summary:**

This paper proposes a method called SoftAD (soft ascent-descent) which aims to improve the "flooding" method. Specifically, it downweights points on the borderline, limits the effects of outliers, and retains the ascent-descent effect of flooding, with no additional computational overhead. Stationarity guarantees of these two methods are provided. Empirical results illustrate that SoftAD can have comparable classification accuracy with flooding while enjoying a smaller loss generalization gap and model norm.

**Strengths:**

1. Overall, this paper is well written.
2. The experimental results demonstrate the merit of the proposed method SoftAD over two baselines (i.e., flooding and SAM). Specifically, SoftAD has competitive classification accuracy with these two methods while enjoying a smaller loss generalization gap and model norm.

**Weaknesses:**

1. The motivation might be confusing. As the authors claimed, there are situations where we are interested in other metrics such as model complexity or average surrogate loss at test time. I can not imagine these situations except the generalization analyses of classification accuracy might involve these factors.

2. The proposed method is heuristic which lacks theoretical support. More specifically, generalization analyses are missing to support the claims.

3. Although it is good to give the optimization convergence guarantees of SoftAD and flooding, they can be not directly comparable because one is high probability bound and another is expectation bound.

**Questions:**

Please give the situations where we are interested in other metrics such as model complexity or average surrogate loss at test time.

**Limitations:**

Please see the Weaknesses part.

---

> ### Author Rebuttal · Authors · 2024-08-02
>
> Thanks to the reviewer for their time in reading the paper and writing the review. Below, we respond to the points raised.
>
> > Please give the situations where we are interested in other metrics such as model complexity or average surrogate loss at test time.
>
> This is an important point, thanks. While there are numerous possibilities, here are a few that have arisen in our own work.
>
> - Training calibrated classifiers (e.g., [Zhao et al. 2021](https://proceedings.neurips.cc/paper/2021/hash/bbc92a647199b832ec90d7cf57074e9e-Abstract.html), [Berta et al. 2024](https://proceedings.mlr.press/v238/berta24a.html)). Using SoftAD to make a small sacrifice to performance at training time without the choppy flooding updates can be quite effective in terms of realizing a good tradeoff between accuracy and calibration at test time, by ensuring the test-time average surrogate loss (cross-entropy, typically) is at a desirable level. The basic principles underlying this point are another big topic; we are working on a separate paper in this direction.
>
> - Learning under fairness constraints, where fairness is measured in terms of dispersion/variance in performance between sensitive sub-populations (e.g., [Hashimoto et al. 2018](https://proceedings.mlr.press/v80/hashimoto18a.html), [Williamson and Menon 2019](https://proceedings.mlr.press/v97/williamson19a.html)). One standard approach is to introduce convex approximations which put more weight on worst-case examples and can approximate the mean + standard deviation of the surrogate losses, such as conditional value-at-risk (CVaR) or distributionally robust optimization (DRO). These methods, however, are extremely sensitive to outliers, and by design discard a lot of data in order to emphasize the worst-case examples. In contrast, SoftAD doesn't discard any data, but instead by design it encourages the base losses to be well-concentrated around $\\theta$, which has a direct impact in terms of fairness-as-dispersion. This is also a big topic on its own, which we are currently working on.
>
>
> > Although it is good to give the optimization convergence guarantees of SoftAD and flooding, they can be not directly comparable because one is high probability bound and another is expectation bound.
>
> Our intention with these two results was to emphasize how with all else equal, using standard analytical techniques, SoftAD can readily be given high-probability guarantees on the test-time objective function of interest, whereas flooding only gets in-expectation guarantees on an approximation of that objective function. In-expectation guarantees can be easily converted into high-probability guarantees using Markov's inequality, but they will be exponentially worse than those for SoftAD in terms of dependence on the confidence level; we can definitely make this explicit in the paper by adding a remark with discussion.
>
>
> > generalization analyses are missing to support the claims.
>
> Please let us emphasize that the "convergence properties" we prove in section 3.2 are not simply convergence guarantees in terms of the objective used at training time, but rather are results given in terms of the population objective (i.e., the objective at test time). In Proposition 2, stationarity is given in terms of the SoftAD risk function at test time defined in equation (10). These are guarantees in terms of the test-time objective for an algorithm run using training-time data, so we have intended them to be formal insights in terms of off-sample generalization. If the reviewer has any additional questions on this point that would help them raise their score or confidence level, we are happy to respond; this is an important point characterizing our formal results.

---

> > ### Comment · Area_Chair_7Xv8 · 2024-08-14
> > **need your reply**
> >
> > To Reviewer JmzP,
> >
> > Please reply to the authors' comments because your score is low compared to the other reviewers.
> > If you are going to keep the current score, please explain in detail why.
> >
> > best
> >
> > Area Chair,

---

### Official Review · Reviewer_55bG · 2024-07-31

**Soundness:** 3
**Presentation:** 3
**Contribution:** 3
**Rating:** 5
**Confidence:** 2

**Summary:**

The paper proposes a new optimization technique, Soft Ascent-Descent (SoftAD), aimed at enhancing the stability and performance of machine learning models. The method balances loss minimization and model complexity, positioning itself as an improvement over existing techniques like Flooding and SAM. The paper provides a theoretical foundation, empirical studies across multiple datasets, and a comparison of convergence properties with existing methods, claiming superior stability and often better test loss outcomes.

**Strengths:**

The paper discusses SoftAD designed to balance minimizing loss and controlling model complexity. This method is compared to existing optimization techniques like Flooding and SAM and is shown to provide better stability and often superior test loss outcomes. The theoretical foundation of SoftAD is robust, with clear mathematical formulations and comparisons to existing methods. Empirical validation includes extensive experiments on multiple datasets, such as CIFAR-10, CIFAR-100, FashionMNIST, and SVHN, demonstrating the effectiveness of SoftAD in practical applications. The paper offers a comprehensive analysis of model norms and loss trajectories, underscoring the practical benefits of the method. Additionally, the document is well-structured, featuring clear explanations, figures, and tables that effectively support its claims and contributions.

**Weaknesses:**

1.	The SoftAD method may be more complex to implement compared to traditional methods, which could limit its adoption. Implementing SoftAD involves managing both ascent and descent phases, which adds to the complexity. This requires additional code for conditionally switching between ascent and descent based on the loss value and the threshold θ.
The complexity is implied by the need to incorporate equations (3) and (4) for the ascent and descent phases into the optimization loop.

2.	The method's performance depends on the setting of the threshold parameter θ, which can be non-trivial to optimize. The threshold θ determines when the algorithm switches between ascent and descent phases, directly impacting the performance. Choosing an inappropriate θ can lead to suboptimal results.

3.	While the experiments are thorough, additional datasets and real-world applications could further validate the robustness of SoftAD. The experiments are conducted on CIFAR-10, CIFAR-100, FashionMNIST, and SVHN. While these datasets are standard benchmarks, they do not cover the full range of possible applications, especially large-scale or domain-specific datasets.
The scope of the experiments is detailed in Section 4, where the datasets used are listed.

4.	The paper does not provide a detailed analysis of how sensitive the performance is to hyperparameters other than θ. Changes in other hyperparameters can significantly affect the performance of optimization algorithms. The focus on θ without detailed analysis of other hyperparameters is evident in Section 5, where the experiments primarily vary θ.
5.	The paper acknowledges the difficulty in setting the threshold θ but does not provide a concrete solution for it, which might be a crucial limitation for practical implementation.
6.	While SoftAD shows better performance in terms of test loss and model norms, it is not always superior in terms of test accuracy when compared to SAM, as noted in the empirical results.

**Questions:**

1.	How do you suggest optimizing the threshold parameter θ in practice? Are there any heuristics or automated methods that can be employed?

2.	How does SoftAD perform on larger datasets and models, such as those used in natural language processing or large-scale image recognition tasks?

3.	Besides θ, are there other critical hyperparameters that significantly affect the performance of SoftAD? How sensitive is the method to these parameters?

4.    Can you provide examples or case studies where SoftAD has been successfully applied in real-world scenarios?

**Limitations:**

Refer to Weaknesses

---

> ### Author Rebuttal · Authors · 2024-08-02
>
> We thank the reviewer for their time in carrying out this review. Below, we respond to the questions raised.
>
> ---
>
> > 1. How do you suggest optimizing the threshold parameter $\\theta$ in practice? Are there any heuristics or automated methods that can be employed?
>
> The ideal setting in practice will inevitably depend on how we want to measure "good performance," but if good performance means a small average value of both surrogate loss and 0-1 loss, then much like the perturbation radius for SAM and the analogous threshold $\\theta$ in Flooding, a small value (usually between 0.01 and 0.25) tends to be a safe bet. In our tests, we used validation accuracy for $\\theta$ selection, and we find that all else equal, more complex tasks (more classes, more extraneous features) tend to lead to a larger value of $\\theta$ being chosen, e.g., larger values for CIFAR-100 and SVHN compared with CIFAR-10 and FashionMNIST.
>
> In terms of possible directions for setting the threshold $\\theta$ in a wide variety of learning settings, one natural approach is to simply run vanilla ERM once, and set $\\theta$ to the mean + standard deviation of the training losses incurred by ERM, noting that even with $\\theta = 0$ (or whatever the minimum possible loss value is), our SoftAD does not collapse into ERM. A modified version of this approach for binary classification, which is slightly more theoretically motivated, is to [incorporate an estimate of the Bayes error](https://openreview.net/forum?id=FZdJQgy05rz), which can then be leveraged to select an early stopping time for the ERM-based approach just mentioned. This Bayes error estimation work comes from the same group that originally proposed Flooding, but it appears they have not integrated the two methods.
>
> > 2. How does SoftAD perform on larger datasets and models, such as those used in natural language processing or large-scale image recognition tasks?
>
> We have not tested SoftAD on NLP datasets yet, but we have run analogous tests using ResNet classifiers trained on a variety of subsets of ImageNet, and the trends observed were essentially the same as those observed in the current manuscript. We can definitely add these results as supplementary material to the current manuscript.
>
> > 3. Besides $\\theta$, are there other critical hyperparameters that significantly affect the performance of SoftAD?
>
> The only hyperparameter inherent to SoftAD as formulated in this paper is $\\theta$. Changes to learning rate, weight decay, and momentum hyperparameters of the SGD optimizer used naturally can impact performance, but in our exploratory analysis the change looks to be essentially uniform across the different methods considered.
>
> > 4. Can you provide examples or case studies where SoftAD has been successfully applied in real-world scenarios?
>
> One recent application that we have been directly involved in is to cybersecurity datasets (e.g., [CIC-MalMem-2022](https://www.unb.ca/cic/datasets/malmem-2022.html)) with a distinct imbalance between benign examples and attack examples, and a lot of diversity within certain types of attacks. We applied SoftAD in combination with distributionally robust optimization (DRO; to put more weight on the tails) to achieve better balanced performance across imbalanced data types, without knowing which example falls into which type of attack, a cost-efficient and type-agnostic alternative to methods like [SharpDRO](https://github.com/zhuohuangai/SharpDRO/tree/main), which we found to work quite effectively (results have not yet been published, in review currently). While specific applications are not the focus of this paper, we can definitely add a remark highlighting the potential practical applications based on our experience thus far.
>
> ---
>
> __Comment regarding code complexity:__
>
> Please let us make a quick remark regarding the following comment raised by the reviewer in the "weaknesses" field.
>
> > Implementing SoftAD involves managing both ascent and descent phases, which adds to the complexity. This requires additional code for conditionally switching between ascent and descent based on the loss value and the threshold $\\theta$.
>
> While admittedly this may not be immediate from the equations in our paper, in fact the implementation of SoftAD is essentially no more complicated than vanilla ERM implemented with SGD. All we do is pass the losses through a wrapper function, which is differentiable and works smoothly with autograd. The switch between ascent and descent is automatic. For anonymity purposes, we have kept our GitHub repository of code anonymous, but here is a simple example of how it would work for training using PyTorch (the `criterion` here corresponds to equation (7) in the paper):
>
> ```
> outputs = model(inputs)
> losses = loss_function(outputs, labels)
> criterion = theta + rho(losses-theta).mean()
> optimizer.zero_grad()
> criterion.backward()
> optimizer.step()
> ```
>
> Apologies for the long response, but we feel the simplicity of implementation is an important point we hope the reviewer can appreciate. At the very least, the implementation is definitely a lot simpler than SAM, and uses half the number of gradients per step. We can definitely add a remark in the paper to ensure this is crystal-clear to readers. If you have any additional questions that would help you raise your score or confidence, please do not hesitate to ask.

---

### Decision · Program_Chairs · 2024-09-25

**Decision:**

Accept (poster)

**Comment:**

The paper introduces SoftAD, a novel optimization method aimed at balancing loss minimization and model complexity control. It is well-supported by a strong theoretical foundation and extensive empirical validation across several datasets, demonstrating its effectiveness compared to existing methods like Flooding and SAM. The paper is well-organized, with clear explanations and visual aids that support its claims.
The authors have carefully and appropriately responded to the reviewers' concerns at the time of submission in the rebuttal phase and have provided reasonable responses. In conclusion, the paper would make a significant contribution to the field and is appropriate for acceptance.